# Benchmarking the Scientific Mind: A Pathology-Derived Biomedical VQA Benchmark for Complex Scientific Reasoning

**Ziyu Zhao** [* 1 2]  **Yiyang Liu** [* 3]  **Yajiao Wang** [* 4]  **Xiaotao Wang** [* 2]
**Yang Li** [5]  **Yuyang Peng** [3]  **Jiaheng Zhou** [2]
**Jinqiao Wang** [2 1 6]  **Yingying Chen** [7]  **Ge Yang** [1 2]  **Haixin Wang** [7]

## Abstract

Despite progress of Multimodal Large Language Models (MLLMs) in biomedical visual question answering (VQA), existing benchmarks provide limited assessment of their scientific reasoning capabilities. Most datasets adopt single-image question construction and outcome-oriented evaluation, where correctness is judged by answer plausibility rather than alignment with experimental evidence. Such formulations fail to capture the evidence-constrained, multi-step nature of biomedical reasoning, and obscure whether models can derive conclusions through causal interpretation of experimental observations.

To address these critical gaps in reasoning evaluation, we propose a principled benchmark construction framework that reconstructs scientific reasoning paths directly from biomedical literature. By jointly modeling clusters of experimentally related images together with their captions and context, the framework generates tightly coupled question–reasoning–answer triples that require multi-image integration and explicit evidence-driven inference. Based on this framework, we introduce **SORBE** (**S**cientific **O**bservation & **R**easoning for **B**iomedical **E**valuation), a large-scale multi-image pathology-derived biomedical VQA benchmark designed to evaluate evidence alignment and multi-step experimental reasoning. Under a process-oriented evaluation metric, state-of-the-art biomedical-specialized MLLMs exhibit substantial performance degradation, revealing systematic limitations in evidence grounding and causal reasoning that are not reflected by existing benchmarks. Data and code are available at: https://github.com/UniverseOfUniverse/SORBE.git.

## 1. Introduction

The rapid adoption of Multimodal Large Language Models (MLLMs) in biomedicine signals a shift from surface-level visual recognition toward scientific reasoning over experimental evidence (Moor et al., 2023; Chen et al., 2024b). Beyond answering what is visible, biomedical Visual Question Answering (VQA) aspires to support hypothesis evaluation, mechanistic interpretation, and evidence-based decision-making. However, despite impressive reported performance, current MLLMs remain poorly assessed on the very capabilities that define scientific intelligence, leading to a growing evaluation crisis (Mahmood, 2025).

A fundamental reason is that most existing biomedical VQA benchmarks conflate **semantic plausibility** with **scientific correctness**. Benchmarks centered on single images and isolated questions (Lau et al., 2018; Liu et al., 2021) primarily reward linguistic coherence or pattern matching, allowing models to succeed through probabilistic associations rather than alignment with experimental evidence. In contrast, scientific questions are intrinsically **evidence-constrained**: answers must be uniquely supported by observable results, and alternative interpretations are invalidated by the data itself. As a result, high benchmark scores often obscure a model's inability to ground its predictions in verifiable visual and scientific evidence, undermining both trust and utility in real scientific settings.

In biomedical fields, particularly in disciplines such as pathology, scientific reasoning is not a single-step infer-

---

[*]Equal contribution  [1]School of Artificial Intelligence, University of Chinese Academy of Sciences, Beijing, China [2]Institute of Automation, Chinese Academy of Sciences, Beijing, China [3]Peking University, Beijing, China [4]National Science Library, University of Chinese Academy of Sciences, Beijing, China [5]Department of Information Resource Management, University of Chinese Academy of Sciences, Beijing, China [6]Wuhan AI Research, Wuhan, China [7]Foundation Model Research Center, Institute of Automation, Chinese Academy of Sciences, Beijing, China. Correspondence to: Haixin Wang <haixin.wang@nlpr.ia.ac.cn>, Ge Yang <ge.yang@ia.ac.cn>, Yingying Chen <yingying.chen@nlpr.ia.ac.cn>.

*Proceedings of the $43^{rd}$ International Conference on Machine Learning*, Seoul, South Korea. PMLR 306, 2026. Copyright 2026 by the author(s).

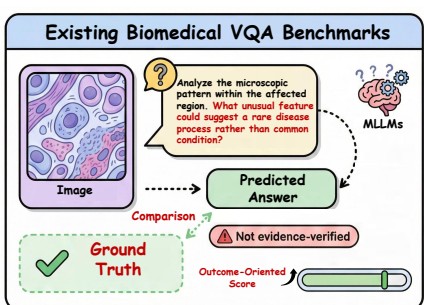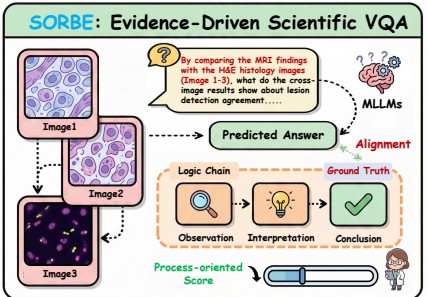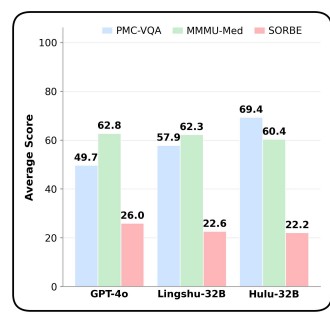

*Figure 1.* **(A)** Comparison between the typical biomedical VQA Benchmark and our **SORBE**. Our process-oriented scoring evaluates verifiable reasoning structure rather than subjective explanation quality. **(B)** Scores of the three models, GPT-4o, Lingshu-32B, and Hulu-32B, on PMC-VQA, MMMU-Med, and our **SORBE**, respectively

ential process. Rather, it typically unfolds as a closed loop procedure that integrates experimental sample preparation, visual observation, comparative analysis, and logical deduction. A question about a microscopy image, for example, is inseparable from the experimental condition it represents, the controls it is compared against, and the causal hypothesis being tested. Existing benchmarks rarely capture this multi-step dependency, reducing scientific reasoning to shallow input–output mappings and preventing evaluation of whether models can internally simulate causal experimental logic rather than merely predict likely answers.

This limitation stems directly from prevailing dataset construction paradigms. Expert-curated benchmarks offer high quality but are inherently small and difficult to scale (Burgess et al., 2025). Automated approaches, on the other hand, typically generate questions by loosely paraphrasing image captions or local descriptions (Zhang et al., 2024; Hu et al., 2023). Such pipelines implicitly assume that semantic relevance suffices for scientific validity, producing questions that can be answered without reconstructing the underlying experimental narrative. This lack of logical continuity not only fosters superficial correlations and linguistic priors (Li et al., 2025), but also exacerbates issues such as hallucinated visual evidence, answer leakage, and dataset bias, ultimately weakening the credibility of benchmark-driven progress (Laskar et al., 2024).

We argue that evaluating scientific VQA requires a principled departure from the single image–single question and outcome-oriented scoring paradigm. A faithful benchmark must instead reflect how scientists reason: integrating multiple related images, grounding observations in context, and traversing explicit chains of evidence to reach a conclusion. In this setting, correctness is defined not by plausibility but by **evidence alignment**, and reasoning quality is determined by whether intermediate conclusions are causally supported by experimental data.

To this end, we propose a principled benchmark construction framework for constructing pathology-derived biomedical VQA benchmarks that explicitly reconstruct scientific reasoning paths from the literature. Our framework transforms clusters of related images, captions, and context into tightly coupled **question–reasoning–conclusion** triples, necessitating multi-step, evidence-driven inference. Specifically, the framework includes three key stages: (1) Contextual Knowledge Extraction, distilling experimental backgrounds; (2) Reasoning Path Construction, building logic chains from visual observation and contextual knowledge; and (3) Question Synthesis, generating high-quality questions and combining the reasoning process and conclusions in the logic chain as a ground-truth reference.

Based on this framework, we construct the **SORBE** (**S**cientific **O**bservation & **R**easoning for **B**iomedical **E**valuation) benchmark, a large-scale multi-image VQA benchmark explicitly designed to evaluate evidence-driven scientific reasoning capabilities in the biomedical field. Unlike existing benchmarks such as MicroVQA (Burgess et al., 2025), PMC-VQA (Zhang et al., 2024), and SLAKE (Liu et al., 2021), which primarily focus on image recognition or description, SORBE requires MLLMs to reconstruct multi-image causal logical reasoning (As illustrated in Figure 1.A). Under our process-oriented evaluation metric, empirical results reveal a stark performance gap, explicitly exposing the limitations of current MLLMs in genuine, multi-step biomedical reasoning (As illustrated in Figure 1.B).

Our contributions are threefold:

(1) We propose a principled benchmark construction framework that transforms biomedical literature into multi-image VQA samples by explicitly reconstructing experimental reasoning paths, integrating images, captions, and context into tightly coupled **question–reasoning–conclusion** triples.

(2) Based on this framework, we present **SORBE**, a

pathology-derived, multi-image biomedical VQA benchmark that shifts the evaluative focus from descriptive image recognition to systematic experimental reasoning. **SORBE** requires MLLMs to align complex visual evidence and execute multi-step logic, thereby probing the boundaries of their causal inference rather than mere semantic pattern matching

(3) We conduct a comprehensive evaluation of state-of-the-art MLLMs on SORBE, revealing systematic failures in evidence grounding and multi-step scientific reasoning that are not captured by existing benchmarks.

## 2. Related Works

### 2.1. Biomedical VQA Benchmarks

Biomedical Visual Question Answering benchmarks have grown in scale and complexity (Table 1). However, most remain focused on perceptual or retrieval tasks, lacking structured reasoning chains.

**Small-Scale, Expert-Curated Benchmarks.** Early benchmarks relied on expert annotation for clinical validity, yielding high-quality but small-scale datasets. For instance, VQA-RAD(Lau et al., 2018) used manually curated radiology images with questions and answers validated by clinical trainees, while MMMU-Med(Yue et al., 2024) had domain-knowledge students collect questions from textbooks. These resources emphasize correctness and relevance, yet remain limited in size, diversity, and anatomical coverage. Cognitively, they correspond mainly to Level 1 (Recall) or Level 2 (Basic Skills) in Webb's Depth of Knowledge (DOK) framework (Webb, 2002), assessing recognition or description without requiring complex evidence-based reasoning.

**Semi-Automated and Expert-Structured Pipelines.** Subsequent benchmarks adopted systematic, pipeline-based methods with structured protocols to scale while preserving expert oversight. For instance, SLAKE (Liu et al., 2021) used an expert-guided pipeline combining physician annotation, a medical knowledge graph, and template-based question generation by doctors. PathVQA (He et al., 2020) employed a semi-automated pipeline mining textbooks, with NLP-based simplification and templating followed by human verification. These approaches produce larger datasets (tens of thousands of samples) but often result in limited linguistic diversity or reasoning depth, as seen in PathVQA's template-driven formulation.

**LLM-augmented Generation and Scalable Curation.** Leveraging large language models (LLMs), recent benchmarks achieve unprecedented scale through automated generation followed by human or model-based filtering. For example, PMC-VQA (Zhang et al., 2024) uses ChatGPT to generate MCQs from PubMed Central captions with auto-

matic filtering, while MedXpertQA-MM (Zuo et al., 2025) and MicroVQA (Burgess et al., 2025) employ GPT-4o for augmentation and rephrasing, incorporating expert verification for quality. OmniMedVQA (Hu et al., 2024) and GMAI-MMBench (Chen et al., 2024a) demonstrate extreme scalability by converting existing medical datasets into VQA format via templates and LLM paraphrasing. While coverage is greatly expanded, questions in such benchmarks often remain focused on classification or attribute retrieval rather than integrative reasoning.

**Towards Multi-Image and Multi-Step Scientific Reasoning.** To address the limitations of single-image tasks, recent benchmarks incorporate comparative or sequential contexts. MIMIC-Diff-VQA (Hu et al., 2023) introduces longitudinal QA from paired chest X-ray studies. MedFrameQA (Yu et al., 2025) advances this by extracting keyframes from medical videos for multi-frame VQA. However, these benchmarks focus on clinical time-series, not the multi-figure, methodological evidence chains typical of research articles.

Our benchmark, **SORBE**, achieves DOK of Level 3 (Strategic Thinking) (Webb, 2002). SORBE demands models use visual evidence to explain "why" or "how"—essential for scientific reasoning. As shown in Table 1, while existing benchmarks contain mostly lower-DOK questions, SORBE provides a structured evaluation of research logic, requiring closed-loop, evidence-based reasoning across multiple experimental images and methodologies.

### 2.2. Process-Oriented Evaluation of Reasoning

Recent work on process-oriented evaluation of large language models (LLMs) aims to assess internal reasoning beyond final-answer accuracy. One approach uses process reward models (PRMs) to supervise intermediate steps (Uesato et al., 2022; Lightman et al., 2024), but these learned rewards are often hard to interpret, making them less suitable as transparent evaluation tools. Other efforts evaluate chain-of-thought faithfulness (Lanham et al., 2023) or use stepwise consistency (Wang et al., 2023), yet these mainly test linguistic coherence rather than grounding in external evidence. MME-CoT (Jiang et al., 2025a) and M3CoTBench (Jiang et al., 2026) further evaluate explicit chain-of-thought reasoning in multimodal and medical image understanding tasks, respectively. However, their evaluation does not target step-level verification against multi-image biomedical experimental evidence chains.

In contrast, our Logic-Coupled Reasoning (LCR) score (Equation 3) formulates evaluation as a structured evidence-verification task, enforcing causal dependencies between observations, interpretations, and conclusions. This approach is especially critical in domains like biomedicine, where hallucinated evidence constitutes a categorical failure. By anchoring evaluation in verifiable evidence rather than sub-

*Table 1.* Overview of Key Biomedical VQA Benchmarks. Our benchmark stands out by integrating multi-image visual signals with complete research logic, moving beyond discrete recognition to holistic scientific reasoning.

| Benchmark | Year | Focus | Multi-image | DOK Level | Reasoning Path Scoring |
|---|---|---|---|---|---|
| VQA-RAD | 2018 | Radiology QA | No | L1 & L2 | No |
| PathVQA | 2020 | Pathology QA images | No | L1 | No |
| SLAKE | 2021 | Medical QA | No | L1 & L2 | No |
| MIMIC-Diff-VQA | 2023 | Differential diagnosis | Yes | L2 | No |
| PMC-VQA | 2023 | Biomedical figures | No | L1 & L2 | No |
| OmniMedVQA | 2024 | Generalist evaluation | No | L1 & L2 | No |
| GMAI-MMBench | 2024 | Clinical VQA | No | L1 | No |
| MMMU-Med | 2024 | Multi-disciplinary QA | No | L1 & L2 | No |
| MicroVQA | 2025 | Scientific hypothesis | No | L3 | No |
| MedXpertQA-MM | 2025 | Expert clinical reasoning | No | L3 | No |
| MedFrameQA | 2025 | Medical video QA | Yes | L3 | No |
| M3CoTBench | 2026 | Medical image CoT | No | L3 | Yes |
| **SORBE (Ours)** | **2026** | **Scientific reasoning** | **Yes** | **L3** | **Yes** |

jective explanation quality, LCR score addresses a gap left by prior metrics.

## 3. Methods

### 3.1. Evidence-Driven Scientific VQA

We formulate biomedical visual question answering as an evidence-driven scientific reasoning challenge, rather than a conventional pattern-matching answer prediction task. Unlike prior benchmarks that implicitly rely on latent reasoning, our formulation makes the reasoning process explicit and externally verifiable. This does not increase reliance on LLM reasoning, but instead exposes and constrains it. This framework shifts the evaluative focus from simple label matching to the validation of multi-step cognitive processes. The core objective of this benchmark is to evaluate whether an MLLM can synthesize these inputs to construct a verifiable and traceable reasoning trajectory, rather than merely arriving at a statistically plausible final answer.

To this end, we explicitly decompose the ground-truth reference into a sequence of structured intermediate reasoning states: **Observation → Interpretation → Sub-Conclusion**. Observation corresponds to a faithful description of visually perceivable patterns in the image (e.g., staining intensity, spatial localization, and structural differences). Interpretation maps these visual cues to biologically meaningful concepts using domain knowledge, such as relative protein expression levels, cell-type specificity, or developmental relevance. Sub-Conclusion summarizes a local scientific judgment supported by a single image or anatomical region. The final conclusion in the ground-truth reference can further verify whether the evaluated MLLMs perform cross-image deduction, aggregating and validating local conclusions to derive a globally consistent answer that is aligned with all available evidence.

Formally, given a set of images $\mathcal{I} = \{I_1, \ldots, I_n\}$ and a question $q$, the reasoning process produces a set of intermediate sub-conclusion $\{h_i\}_{i=1}^n$, where

$$h_i = f_{\text{int}}\left(f_{\text{obs}}(I_i), q\right), \tag{1}$$

and the final answer $a$ is obtained via an evidence-aligned aggregation function

$$a = g\left(h_1, \ldots, h_n\right). \tag{2}$$

This Evidence-Driven Scientific VQA setting prioritizes both the accuracy of the final output and the interpretability of the underlying reasoning. It provides a robust metric for benchmarking the next generation of biomedical MLLMs in fields demanding high precision, such as pathology, developmental biology, and molecular mechanism analysis.

### 3.2. Logic-Coupled Reasoning (LCR) Score

To rigorously evaluate multi-step reasoning in biomedical experimental settings, where correctness alone is insufficient without valid scientific justification, we introduce **the Logic-Coupled Reasoning (LCR) Score**, denoted as $S_{\text{LCR}}$, for scoring open-ended answers produced by MLLMs.

Formally, $S_{\text{LCR}}$ is defined as the harmonic mean of a **Conclusion Score** $S_{\text{conc}}$ and a **Reasoning Process Score** $S_{\text{proc}}$:

$$S_{\text{LCR}} = \frac{2 \cdot S_{\text{proc}} \cdot S_{\text{conc}}}{S_{\text{proc}} + S_{\text{conc}}}. \tag{3}$$

The harmonic formulation ensures that high performance can only be achieved when both the final conclusion and the underlying reasoning are valid, preventing degenerate solutions that exploit superficial correctness. $S_{\text{LCR}}$ explicitly couples final conclusion validity with reasoning process faithfulness, reflecting the epistemic requirements of biomedical reasoning.

We apply LLM evaluator as a constrained verifier rather than a free-form judge. It does not generate alternative reasoning,

revise intermediate steps, or access the final answer. This setting aligns with recent uses of LLMs as structural verifiers under explicit logical constraints.

**Conclusion Score.** The conclusion score evaluates the correctness and completeness of the final conclusion relative to the ground-truth reference. The evaluation process first assigns a raw score $S_{\text{raw}} \in [0, T]$, where $T$ denotes the maximum achievable points. This raw score is then normalized to the unit interval as follows:

$$S_{\text{conc}} = \frac{S_{\text{raw}}}{T}. \tag{4}$$

Partial credit is assigned when conclusions are directionally correct but remain incomplete or weakly justified.

**Reasoning Process Score.** To assess reasoning fidelity, we decompose each response into an alignment task against a reference reasoning graph consisting of $N$ experimental branches. Each branch corresponds to an independent line of experimental evidence and is evaluated via three ordered logical nodes:

$$\langle \phi_i, \chi_i, \psi_i \rangle \in \{0, 1\}^3, \tag{5}$$

where

- Observation Correctness ($\phi_i$) captures whether the model correctly identifies task-relevant visual evidence;

- Interpretation Correctness ($\chi_i$) evaluates whether the visual observation is explained using scientifically valid mechanisms;

- Sub-Conclusion Correctness ($\psi_i$) assesses whether a logically necessary intermediate conclusion is derived from the interpretation and can serve as an antecedent to the final conclusion.

We intentionally adopt binary node-level indicators to emphasize failure localization rather than fine-grained partial credit. In evidence-driven biomedical reasoning, missing or hallucinated evidence constitutes a categorical error rather than a semantic judgment.

To explicitly discourage language-based shortcuts and post-hoc rationalization, we enforce causal dependency via cascading penalties: failure at any prerequisite node nullifies all downstream scores within the same branch. The score for the $i$-th experimental branch is defined as:

$$S_{\text{exp}_i} = \frac{\phi_i + \phi_i \cdot \chi_i + \phi_i \cdot \chi_i \cdot \psi_i}{3}. \tag{6}$$

The overall reasoning process score is computed as the average across all experimental branches, Edge cases are handled as described in Appendix G.:

$$S_{\text{proc}} = \frac{1}{N} \sum_{i=1}^{N} S_{\text{exp}_i}. \tag{7}$$

This formulation explicitly encodes the hierarchical dependency between observation, interpretation, and deduction, thereby penalizing hallucinated or logically incoherent reasoning even when the final answer appears correct.

### 3.3. Reasoning Path Construction

We propose an autonomous framework to bridge the gap between high-level scientific reasoning and low-level visual perception by synergistically combining robust visual evidence extraction with structured logical inference for constructing structured reasoning paths from biomedical literature. Our approach addresses the dual challenges of visual hallucination and reasoning drift by enforcing a rigorous "observe-then-reason" pipeline. Specifically, the framework first employs a **Multi-Expert Visual Integration** module to synthesize a consensus-based, hallucination-free observation of images. This structured visual evidence is then fed into a **Logic Chain Generation** module, which constrains the model's cognitive trajectory through a hierarchical reasoning scaffold. By explicitly decoupling visual perception from logical deduction and requiring traceable evidence attribution, our framework ensures that complex scientific conclusions are grounded in verifiable multimodal observations rather than stochastic associations.

**Multi-Expert Visual Integration.** To address the random hallucinations in individual MLLMs, our framework approaches visual evidence extraction through a consensus-driven process using four complementary expert models. By leveraging their diverse strengths in biomedical semantics and general object recognition, the framework generates parallel, independent descriptions. This integration produces a comprehensive set of features that effectively reduces single-architecture bias. We further implement a rigorous inter-agent consensus protocol to filter noise: (i) **Fact Verification** employs majority voting to retain only cross-validated morphological structures; (ii) **Consistency Check** integrates unique expert observations only if they align with fundamental biological principles; and (iii) **Contextual Refinement** prunes irrelevant content against the experimental context. This multi-layered validation ensures that the resulting visual evidence is both structurally consistent and scientifically grounded.

**Logic Chain Generation.** This step decomposes complex reasoning embedded in unstructured scientific corpora into standardized and modular steps, formalized as structured

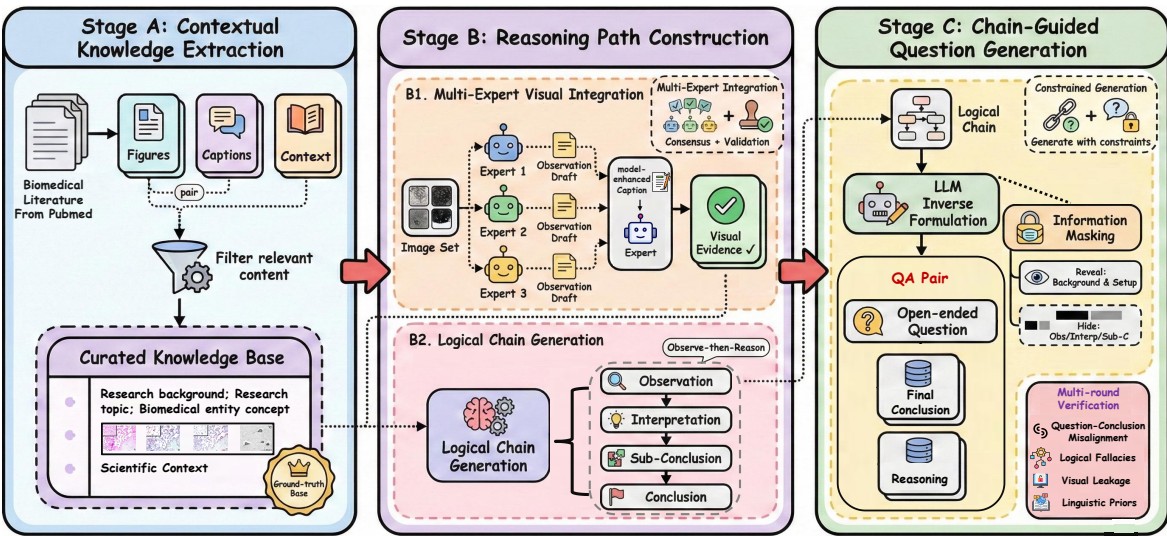

*Figure 2.* **Overview of Benchmark Construction Framework.** **(A)** Contextual Knowledge Extraction, which distills structured experimental metadata from unstructured biomedical literature; **(B)** Reasoning Path Construction, which extracts visual evidence and constructs logic chains from images and contextual knowledge; and **(C)** Question Generation & Filtering, which produces open-ended questions and filters out low-quality questions.

logic chains. We require the large language model (LLM) to organize its responses following a high-level reasoning scaffold of research background → experiments → conclusions. Each experiment is further decomposed into a fine-grained micro-chain consisting of setup → objective → observation → interpretation → sub-conclusion. This hierarchical design explicitly constrains the reasoning trajectory, mitigating common failure modes of LLMs in long-horizon reasoning, such as logical leaps, topic drift, and hallucinations, while improving coherence and interpretability. Crucially, our framework enforces a strict separation between textual context and visual evidence extracted from preceding steps, accompanied by explicit rules governing their integration. During reasoning, the LLM is required to annotate the provenance of each visual observation, specifying whether it originates from a particular image, a textual description, or is missing altogether. By making evidence attribution explicit within the reasoning chain, the model is discouraged from speculative inference and instead guided toward evidence-grounded reasoning.

### 3.4. Benchmark Construction Framework

As illustrated in Figure 2, our framework follows a principled pipeline that transforms unstructured multimodal data into coupled question–reasoning–conclusion triples. We randomly sampled 10% of the final generated dataset for manual validation by biology experts or PhDs. The expert validation confirmed that 93.4% of the samples were of high quality.

**Stage A: Contextual Scientific Knowledge Extraction.**

The process begins by constructing a domain-specific knowledge base from biomedical literature. First, automated tools pair images with their captions and extract textual context. A rigorous filtering step then selects the relevant biomedical content. Finally, core experimental metadata, which includes the experiment background and key biomedical concepts, is distilled into a structured format. This curated knowledge base provides a ground-truth foundation for all subsequent tasks.

**Stage B: Reasoning Path Construction.** The process synergistically integrates robust visual evidence extraction with structured logical inference to construct logic chains. The framework first applies a Multi-Expert Visual Integration module, which synthesizes consensus-based, hallucination-free observations from images through a validation protocol. The resulting visual evidence is then processed by a logic chain generation module, which decomposes scientific reasoning in the knowledge base into hierarchical, standardized steps following a strict observe-then-reason scaffold.

**Stage C: Logic Chain-Guided Question Generation.** The process systematically guides LLMs through a constrained generation pipeline to produce open-ended questions that target complex cognitive reasoning. The process begins with an inverse formulation of questions, whereby the LLM derives questions from a pre-constructed logic chain to ensure the semantic and logical equivalence of the target. An information-masking strategy is employed to define the scope of information in the generated questions, intentionally withholding intermediary reasoning steps (e.g., observations, interpretations, and sub-conclusions) while disclosing

*Table 2.* Comparison with state-of-the-art models on general medical VQA benchmarks and reasoning tasks. **SORBE** is our proposed benchmark. The best results in each group are highlighted in **bold**.

| Models | Standard Medical VQA Benchmarks | | | | | Reasoning & Knowledge | | Ours |
|---|---|---|---|---|---|---|---|---|
| | OMVQA | PMC-VQA | PathVQA | Slake | VQA-RAD | MedXQA | MMMU-Med | SORBE |
| *Proprietary Models* | | | | | | | | |
| GPT-4o | 67.5 | 49.7 | 55.5 | 71.2 | 61.0 | 44.3 | 62.8 | 26.0 |
| Claude Sonnet 4 | 65.5 | 54.4 | 54.2 | 70.6 | 67.6 | 43.3 | 74.6 | 31.8 |
| Gemini-2.5-Flash | 71.0 | 55.4 | 55.4 | 75.8 | 68.5 | **52.8** | **76.9** | **42.0** |
| *Open-source Models (<10B)* | | | | | | | | |
| Lingshu-7B | 82.9 | 56.3 | 61.9 | 83.1 | 67.9 | 26.7 | 54.0 | 19.2 |
| Hulu-Med-7B | 84.2 | 66.8 | 65.6 | 86.8 | 78.0 | 29.0 | 51.4 | 18.2 |
| *Open-source Models (>10B)* | | | | | | | | |
| Qwen2.5-VL-72B | 66.6 | 55.8 | 66.0 | 78.0 | 68.4 | 23.4 | 70.8 | 25.3 |
| Lingshu-32B | 83.4 | 57.9 | 65.5 | 86.7 | 76.7 | 30.9 | 62.3 | 22.6 |
| Hulu-Med-32B | 84.6 | 69.4 | 67.3 | 85.7 | **81.4** | 34.0 | 60.4 | 22.2 |
| Fleming-VL-38B | **87.9** | **76.5** | **68.0** | **89.8** | 76.6 | - | - | 17.8 |

the experimental background and setup. These hidden logic chains and the final conclusion then serve as the ground-truth reference for model benchmarking. To ensure the rationality of the questions and the correctness of the generated logical chains, we conducted multiple rounds of automated quality checks on each generated sample. These checks included: traversing the logical chains to verify alignment between questions and conclusions; examining whether visual leakage occurred in the questions; and assessing whether the LLM predicted correct conclusions based solely on textual information. Low-quality samples were subsequently filtered out.

### 3.5. Benchmark Instantiation: SORBE

The benchmark **SORBE** is constructed based on our framework (in Section 3.4). We initially extracted biomedical multimodal scientific contexts from image-caption pairs in the PathCap (Sun et al., 2024) dataset. To conserve test time and computing resources, 10,000 raw data points were randomly selected for benchmark construction. To ensure the task required complex cross-image synthesis, we filtered out entries containing only a single image, ultimately obtaining 5,867 multi-image samples. We employed an ensemble of four distinct MLLMs (Qwen3-VL, Lingshu-32B, Hulu-Med-32B, and Fleming-VL-38B) to extract fine-grained visual observations. The outputs were then integrated and refined using the Qwen3-235B-Instruct model to suppress random hallucinations. Subsequently, this model was used to generate logic chains and construct question-reasoning-conclusion triples. Finally, DeepSeek-V3.2 was used to implement a rigorous quality control (QC) protocol to filter out samples containing logical fallacies, unreasonable questions, visual leakage, and linguistic priors, generating

3,215 high-quality QA pairs. The entire process was completed within 32 hours using 8 NVIDIA H20 GPUs, with all large-scale model inference executed in parallel, demonstrating the framework's scalability for building large-scale benchmarks. Representative example questions illustrating SORBE's capability coverage and an example instance are provided in Appendix C and Appendix B, respectively.

## 4. Experiments

### 4.1. Experimental Setup

*Table 3.* Fine-grained scores of models on the SORBE benchmark. $Avg.S_{\text{proc}}$, $Avg.S_{\text{conc}}$ and $Avg.S_{\text{LCR}}$ mean the average of all scores $S_{\text{LCR}}$, $S_{\text{proc}}$ and $S_{\text{conc}}$ in SORBE.

| Models | $Avg.S_{LCR}$ | $Avg.S_{proc}$ | $Avg.S_{conc}$ |
|---|---|---|---|
| GPT-4o | 26.0 | 25.8 | 52.3 |
| Claude Sonnet 4 | 31.8 | 32.3 | 55.6 |
| Gemini-2.5-Flash | **42.0** | **42.8** | **61.7** |
| Qwen2.5-VL-72B | 25.3 | 26.4 | 49.1 |
| Qwen3-VL-235B | 40.9 | 42.0 | 60.8 |
| Lingshu-7B | 19.2 | 20.3 | 42.3 |
| Lingshu-32B | 22.6 | 23.1 | 47.7 |
| Hulu-Med-7B | 18.2 | 19.4 | 40.4 |
| Hulu-Med-32B | 22.2 | 22.4 | 46.6 |
| Fleming-VL-38B | 17.8 | 18.4 | 39.8 |

We evaluate several representative MLLMs on **SORBE** to ensure a holistic performance assessment, including **Closed-Source Models:** GPT-4o (Hurst et al., 2024), Gemini-2.5-Flash (Comanici et al., 2025), and Claude 4 Sonnet (Anthropic, 2025); **Open-Source Models:** Qwen3-VL-235B (Bai et al., 2025a); Qwen2.5-VL-72B (Bai et al., 2025b); **Domain-Specific Expert Models:** Hulu-Med

*Table 4.* Ablation study of different scoring metrics across multiple MLLMs. All scores are normalized to [0, 100].

| Metric | GPT-4o | Claude Sonnet 4 | Gemini-2.5-Flash | Lingshu-32B | Hulu-Med-32B | Fleming-VL-38B |
|---|---|---|---|---|---|---|
| **FFJ** | 60.0 | 63.0 | 68.3 | 55.2 | 53.9 | 45.8 |
| **LCR** | 26.0 | 31.8 | 42.0 | 22.6 | 22.2 | 17.8 |

(7B/32B) (Jiang et al., 2025b), Fleming-VL (38B) (Shu et al., 2025), and Lingshu(7B/32B) (LASA Team et al., 2025). We report the LCR scores of different MLLMs on **SORBE** and compare them with other benchmarks. Complete LLM Prompts and evaluation scripts are provided in the Appendix A.

### 4.2. Main Results

**Performance on SORBE.** As shown in Table 2, all evaluated models show significantly lower performance on SORBE compared to traditional medical VQA benchmarks. The proprietary models, particularly Gemini-2.5-Flash, demonstrate superior capabilities with a score of 42.0, while even the best-performing open-source model, Qwen2.5-VL-72B, achieves only 25.3.

The most striking observation is the dissociation between performance on standard medical benchmarks and complex reasoning tasks. For instance, Fleming-VL-38B achieves state-of-the-art results on OMVQA (87.9) and Slake (89.8) but performs poorly on SORBE (17.8). This indicates that SORBE assesses fundamentally different cognitive capabilities beyond traditional medical question-answering.

It is noted that: 1) Performance Gap: Even top-performing models on standard medical VQA benchmarks struggle with SORBE's complex reasoning tasks. 2) Larger open-source models ($> 10$B parameters) generally outperform smaller counterparts but still lag behind proprietary models. 3) Surprisingly, domain-specific biomedical models (Hulu-Med, Fleming-VL) don't outperform generalist models on SORBE, suggesting the benchmark assesses different capabilities than traditional medical VQA tasks like PMC-VQA. The results demonstrate that SORBE presents a challenging testbed requiring advanced multimodal reasoning capabilities beyond standard medical question-answering.

### 4.3. Results Analysis

To better understand model capabilities in complex multi-image reasoning, we compared the process score ($S_{proc}$), conclusion score ($S_{conc}$), and LCR ($S_{LCR}$).

Table 3 reveals several critical patterns: 1) Consistent performance ranking: The relative ranking of models remains consistent across all three metrics, with Gemini-2.5-Flash consistently outperforming other models. 2) LCR as the most challenging metric: All models achieve significantly lower scores on $S_{LCR}$ compared to $S_{proc}$ and $S_{conc}$. For example, GPT-4o achieves 25.8 on $S_{proc}$ and 26.0 on

$S_{LCR}$, despite a much higher 52.3 on $S_{conc}$, indicating that maintaining both logical coherence and correct conclusions is substantially more difficult than achieving either alone. 3) Proprietary models excel in logical coherence: The gap between proprietary and open-source models is most pronounced in $S_{LCR}$, suggesting that proprietary models have better capabilities in maintaining consistent logical chains throughout complex reasoning processes.

We conducted a detailed analysis of three representative models: GPT-4o, Lingshu-32B, and Hulu-32B in Appendix D. Furthermore, a detailed qualitative error analysis of four representative cases is presented in Appendix F.

### 4.4. Ablation Study of Evaluation Metric

In order to evaluate the effectiveness and priority of the LCR metric, this study compares LCR with the naive **Free-Form Judge** ($S_{\text{FFJ}}$) where the LLM-as-a-judge compares the model's answer against generated reference answer text rather than the logic chain and directly assigns the final score. The LLM evaluator employs DeepSeek-V3.2. We assess several MLLMs on the SORBE benchmark to examine how distinct scoring methodologies influence their performance test results.

**Experimental Results and Analysis.** Experimental results in Table 4 demonstrate that LCR yields consistently lower scores than FFJ across all models, indicating a more stringent evaluation of logical consistency. By penalizing "right-answer-wrong-reasoning" cases, LCR effectively mitigates overestimation inherent in naive LLM-as-a-judge scoring. Furthermore, LCR increases the performance margin between top-tier and baseline MLLMs, establishing a higher-fidelity benchmark for rigorous reasoning assessment.

A comparison between LCR and FFJ methods on a representative case is presented in Appendix H.

## 5. Conclusion

**Limitation.** 1) Due to the direct utilization of the biomedical image-caption pair PathCap dataset as the source data, instead of extracting all images from Pubmed literature, experimental/statistical charts are excluded, and image quality is inevitably limited. 2) The strict filtering strategies de-

signed in data preprocessing result in a trade-off regarding data utilization, as literature documents with minor imperfections are inadvertently excluded.

**Summary.** The paper introduces SORBE, a pathology-derived, multi-image biomedical VQA benchmark designed to evaluate evidence-aligned, multi-step scientific reasoning reconstructed from biomedical literature. It proposes a benchmark construction framework that produces question–reasoning–conclusion triples and a process-oriented LCR metric that jointly scores conclusion correctness and reasoning fidelity via hierarchical penalties. Our experiments indicate that state-of-the-art MLLMs degrade substantially under SORBE compared to other outcome-oriented benchmarks, suggesting current systems struggle with evidence-grounded causal reasoning.

## Impact Statement

This paper presents work whose goal is to advance the application of machine learning in the biomedical field. There are many potential societal consequences of our work, none which we feel must be specifically highlighted here.

## Acknowledgments

This work was supported by the National Key Research and Development Program of China under Grant No. 2023ZD0120400 and Grant No. 2024YFF0729202 to G.Y.

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

# A. Complete Prompt Library

In this section, we provide the specific prompt templates used in each stage of our framework.

## A.1. Data Curation and Pre-processing

### A.1.1. BIOMEDICAL CONTENT FILTERING

This prompt is used to filter out non-biomedical data samples.

> **Biomedical Check Prompt**
>
> ```
> You are a data classifier. Your task is to determine if the provided text context
>     describes biomedical, medical, or clinical content (e.g. pathology, anatomy,
>     cell biology, medical imaging, clinical reports).
>
> Input Context:
> {context}
>
> Output Requirement:
> Return ONLY a JSON object with a single boolean field 'is_biomedical'.
> Example: {{"is_biomedical": true}} or {{"is_biomedical": false}}
> ```

### A.1.2. BACKGROUND DISTILLATION

Used to condense long background texts into concise scientific summaries.

> **Background Distillation Prompt**
>
> ```
> As an expert biomedically scientific editor, your task is to distill the provided
>     [Background] text into a concise, focusing on biomedical entity information.
>
> Input:
> [Background]:
> {back_info}
>
> The distilled summary MUST meet the following requirements:
> 1.  Length: a reasonable summary between 100 and 200 words.
> 2.  Focus: Focus on explaining the core scientific problems within the background
>     context, including the important knowledge related to them.
> 3.  Style: Use formal, clear, and objective scientific language.
>
> Output:
> ```

### A.1.3. THEME AND CONTEXT EXTRACTION

Used to identify experimental settings and image themes.

> **Theme & Background Extraction Prompt**
>
> ```
> You are an expert biomedical researcher. Analyze the provided [Context],
>     [Background Info], and [Image Captions] to extract the core subject for EACH
>     image and the overall experimental context.
>
> # Input Data
> [Background]:
> {back_info}
> ```

```
[Context]:
{context}

[Image Captions]:
{captions}

# Task
1. **Experimental background**: Summarize the overall experimental background, or
   setup in **2 sentences or fewer**.

2. **Image Settings**: Describe the specific setup, stage, or setting for each
   image in a single phrase or sentence.

# Output Format
Return ONLY a valid JSON object in the following exact format:
{{
    "Experimental background": "Overall background summary...",
    "Image Settings": {{
        "Image 1": "...",
        "Image 2": "..."
        ...
    }}
}}
```

## A.2. Domain Keyword Extraction

Extracts specific biological entities and categories to guide downstream experts. We categorize the research domains into four main classes as shown in Table 5.

| Category | Description & Keywords |
|---|---|
| **Basic Medical Science** | Focuses on the fundamental mechanisms of life and disease. 
 *Keywords:* Molecular biology, genetics, biochemistry, immunology, physiology, anatomy, neurosciences, cellular pathways, pathogenesis models |
| **Clinical Medicine** | Focuses on the diagnosis, treatment, and management of human diseases. 
 *Keywords:* Specific diseases (e.g., Heart Disease), surgical procedures, patient case studies, treatment outcomes, clinical neurology, ophthalmology, urology, orthopaedics |
| **Diagnostics & Laboratory Medicine** | Focuses on the methods and technologies used to detect and diagnose diseases. 
 *Keywords:* Pathology, histopathology, cytopathology, medical imaging (Radiology, MRI, CT), biomarkers, lab tests, assay development, forensic analysis |
| **Pharmacy & Therapeutics** | Focuses on the discovery, development, and application of drugs. 
 *Keywords:* Pharmacology, drug synthesis, medicinal chemistry, drug targets, therapeutic strategies, drug resistance, clinical trials, pharmaceutical sciences |

*Table 5.* Domain Classification Framework and Keywords

---

**Keyword Extraction Prompt**

```
You are a top-tier biomedical research analyst, skilled at structured information
    extraction and thematic classification. Your task is to perform a two-step
    analysis on the provided [Context] and [Image_caption].

I. INPUT DATA

[Context]:
{context}
*(Note: The [Context] may contain '[Image]' tokens indicating image positions. You
    CANNOT see these images; you MUST rely *only* on the Observation for all visual
    details.)*

[Image_caption]:
{image_caption}

II. STEP 1: THEMATIC CLASSIFICATION
Analyze the content and select **ONE** Main Category (1-4) that best describes the
    core research domain. Use the professional framework below.

CLASSIFICATION FRAMEWORK:
- Basic Medical Science :
  Focuses on the fundamental mechanisms of life and disease.
   (Keywords: Molecular biology, genetics, biochemistry, immunology, physiology,
    anatomy, neurosciences, cellular pathways, pathogenesis models).
- Clinical Medicine :
  Focuses on the diagnosis, treatment, and management of human diseases in patients.
   (Keywords: Specific diseases (e.g., Heart Disease, Endocarditis), surgical
    procedures, patient case studies, treatment outcomes, clinical neurology,
    ophthalmology, urology, orthopaedics).
- Diagnostics & Laboratory Medicine :
  Focuses on the methods and technologies used to detect and diagnose diseases.
   (Keywords: Pathology, histopathology, cytopathology, medical imaging (Radiology,
    MRI, CT), biomarkers, lab tests, assay development, neuropathology, forensic
    analysis, electrocardiography).
- Pharmacy & Therapeutics :
  Focuses on the discovery, development, and application of drugs.
```

```
    (Keywords: Pharmacology, drug synthesis, medicinal chemistry, drug targets,
     therapeutic strategies, drug resistance, clinical trials for drugs,
     pharmaceutical sciences).

III. STEP 2: THEME-GUIDED KEYWORD EXTRACTION
Based on the [Context] and the provided Observation and Interpretation, as well as
    the classification result from STEP 1, extract a list of 10-15 highly specific
    biological or medical keywords.
- **CRITICAL:** Ensure keywords are directly relevant to the selected Main Category.
- **Focus on:** Specific protein/gene names, cell types/morphologies, disease
    names, diagnostic criteria (e.g., grading), specific drugs, or key experimental
    findings.
- **Avoid:** Generic words or phrases.

IV. REQUIRED OUTPUT FORMAT
Output *only* the result in the following exact structure:
[Main Category Name]: keyword1, keyword2, keyword3, ..., keyword15
```

## A.3. Multi-Expert Visual Perception

### A.3.1. SINGLE EXPERT VISUAL DESCRIPTION

The base prompt used by individual MLLMs (e.g., Qwen-VL, etc.) to generate initial captions.

```
Single MLLM Description Prompt

You are an expert biologist and biomedical researcher. You will be given an image
    and [Image_Caption]. Your task is to describe the visual content of the image.

# Input Data
[Image_Caption]:
{caption}

# Task
Provide A detailed description of the visual features present in the image,
    grounded in the [Image_Caption]. Avoid using any conclusive statements.
    Focusing on the observation of visual features in biomedical images.

# Constraints
- Do NOT output a list.
- Do NOT mention "Image 1" or other image indices.
- Output ONLY the description paragraph.
```

### A.3.2. EXPERT CONSENSUS AND AGGREGATION

The core mechanism for merging multiple expert observations into a coherent report.

```
Consensus Integration Prompt

You are a senior biomedical image analyst. You are a senior biomedical image
    analyst. You will receive observations from four different biomedical experts
    regarding the same biomedical image. These observations may include their
    interpretations or inferences based on the image, which you should disregard.

#Task:
You should limit yourself to purely visual descriptions, avoid adding any
    explanatory logic, and extract the common visual information from these
    observations, while avoiding contradictions and ensuring the information is
    biomedically right. Generate a highly accurate "comprehensive observation
    report.

## Bad example: It combines Qwenvl's incorrect "cytoplasmic" description with a
    correct "prominent nuclear staining" description later, resulting in a
    confusing and anatomically impossible description for a single stain.

# Input Observations:

[Model: Fleming]:
{desc_fleming}

[Model: Hulu]:
{desc_hulu}

[Model: Lingshu]:
{desc_lingshu}

[Model: QwenVL]:
{desc_qwenvl}
```

```
# Key Requirements:
1. Voting & Merging Strategy:
   - For overlapping features mentioned by multiple models, use **majority
   voting** to establish the **corroborated facts**.
   - For distinct/unique details mentioned by only one model, **naturally merge**
   them into the description to enrich detail, provided they DO NOT contradict the
   **corroborated facts** or biomedical logic.

2. Pure Observation: Describe ONLY the visible morphological features ( e.g.,
   cells, staining, structures   ). Do not include any reasoning, such as
   "consistent with...", "indicates...", "seems to...", "suggests some
   expression...", "may represent...".  Focus on the visual features of the image
   itself; do not draw conclusions or inferences based on interpretations or
   deductions from the image.
3. Integration: Output a single, coherent paragraph merging the corroborated facts
   and valid, unique details naturally.

# Output
(Output ONLY the Integration description paragraph.)
```

### A.3.3. CONTEXT-ENHANCED CAPTIONING

Synthesizes visual observations with background knowledge and keywords.

---

**Context-Enhanced Captioning Prompt**

```
You are an expert biologist and biomedical researcher. You will be given [Context],
    [Background], [Keywords], and a set of initial [Image_captions].

# Your goal is to generate a "Context_Enhanced_Captions" object. You must process
    the data in two distinct steps for each image: Verification ([Observation]) and
    Analysis ([Interpretation]).

## Verification ([Observation]): Rigorously validate the [Image_captions] to
    correct only factual errors based on [Context] while strictly preserving all
    non-conflicting visual details, ensuring the output remains a purely
    descriptive report devoid of any explanatory logic.

## Analysis ([Interpretation]): Synthesize the verified visual observations with
    [Context], [Background], and [Keywords] to explain the deep biological
    principles, mechanisms, or pathologies underlying the visual data.

# Input Data

[Background]:
{distilled_background}

[Keywords]:
{keywords}

[Context]:
{context}

[Image_captions]:
{vl_captions_json}

# Task Guidelines & Logic

# 1. Alignment Strategy
```

---

The [Context] text contains `[Image]` tokens (e.g., [Image 1], [Image 2]). These
    tokens mark the exact location where the image is discussed.
You must use the text immediately surrounding these `[Image]` tokens to verify the
    identity and features of the corresponding image in [Image_captions].

# 2. Field: "observations" (Strict Visual Verification)
Goal: Correct the [Image_captions] ONLY if they are factually wrong based on the
    [Context], while preserving correct visual details.
## Minimal Modification Rule: Do not rewrite the caption if it is consistent with
    the text. Only edit specific words or phrases that contradict the [Context].
### Correction Protocol:
    * If [Image_captions] says "blue stain" but [Context] specifies "red stain",
    change it to "red stain".
    * If [Image_captions] mentions a visual detail (e.g., "irregular shape") that
    is NOT mentioned in [Context], PRESERVE IT. Do not delete valid visual details
    just because the text doesn't mention them.
### Anti-Hallucination Rule**: Do NOT add biological reasoning, causal
    relationships, or background knowledge into this field. Keep it purely
    descriptive (shapes, colors, positions ).

# 3. Field: "interpretations" (Biological Reasoning)
Goal: Explain the biological significance of the verified [Observation] using
    [Background] and [Context].
* Use this field to bridge the gap between "what we see" ([Observation]) and "what
    it means" (Context).
* Explain the function, process, or pathology visible in the image.
* Synthesize information from the [Background] to provide depth.

# 4. Summary Generation
## [Observation] summary: If there is only one image, provide a concise visual
    overview of that specific image. If there are multiple images, synthesize the
    common visual themes across all panels. MUST remain purely descriptive (no
    reasoning).
## [Interpretation] summary: If there is only one image, provide a final biological
    conclusion or diagnosis based on the analysis of the single image. If there are
    multiple images, provide a joint analysis of the overall biological conclusion
    derived from the combination of these images.

# Output Format
Provide the final answer as a JSON object with a single root key
    "Context_Enhanced_Captions".
The output must strictly separate visual descriptions from analytical insights.
Ensure the output is a valid JSON list of strings within the structure,
    corresponding one-to-one with the original captions.

```json
{{
  "Context_Enhanced_Captions": {{
    "observations": {{
      "Image 1": "...",
      "Image 2": "...",
      ...
      "summary": "..."
    }},
    "interpretations": {{
      "Image 1": "...",
      "Image 2": "...",
      ...
      "summary": "..."
    }}
  }}
}}
```

```
# Key Requirements
1. JSON Validity: The output must be directly parseable by json.loads.

2. Separation of Concerns:
"[Observation]" = Pure Vision + Contextual Correction (No "because...", No
    "indicating that...").
"[Interpretation]" = Vision + Contextual Logic (Explain the "Why").
3. Context Fidelity: Do not hallucinate details not present in the image or the
    text.
4. Count Match: The number of keys in the dictionary must match the number of input
    images.
5. Keyword Integration: Utilize the [Keywords] to anchor your terminology in
    "[Interpretation]", ensuring the specific modality, staining technique, or
    pathological classification is accurately reflected.

Generate [Context-Enhanced Captions]:
```

## A.4. Reasoning and QA Generation

### A.4.1. VISUAL ELEMENT QA GENERATION

Generates vision-centric QA pairs to test low-level perception.

---

**Visual QA Generation Prompt**

```
You are an expert in biomedical image analysis. Your task is to generate
    vision-centered [Question]-[Answer] pairs based ONLY on the provided
    [Observation]. Extract a list of all unique biomedical entities (e.g., cell
    types, staining, anatomical structures) mentioned in the [Observation].

# Input Data
[Observation] refers to objective visual descriptions of each image, and the
    summary consolidates the visual findings to provide a holistic overview, the
    inter-image relationship across the images
{Observation}

# Output

Generate [Question]-[Answer] pairs that asks for the specific biomedical visual
    features mentioned in the [Observation]. If there is only one image available,
    then only use that single image for the question. If there are multiple images,
    selecting several (but not necessarily all) closely related images can generate
    reasonable questions.
The goal is to verify if the model can "see" the low-level details before
    performing high-level reasoning.

## Key Requirements (MUST FOLLOW):
1. Strictly Visual: The [Question] MUST focus ONLY on visual attributes. Aim for
    high diversity in questions suitable for multi-image biomedical analysis.
IF [Observation] contains Only One Image**: You MUST generate a Descriptive
    question specific to that image (e.g., "Describe the staining intensity of the
    cytoplasm in Image 1."). Do NOT hallucinate other images.

IF [Observation] contains multiple images, you can generate questions about the
    relationships between these images.

Examples include:
Comparative Morphology: 'Compare the nuclear irregularity observed in different
    images.'
Feature Characterization: 'Describe the texture and staining intensity of the
    cytoplasm in...'
Structural Architecture: 'How does the arrangement of inflammatory cells differ
    between different images?'

2. No Interpretation: The [Question] and [Answer] MUST NOT contain diagnostic
    conclusions, biological significance, or "Why" reasoning. Do not use words like
    "suggests", "indicates", or "diagnosis".
3. Image Reference in [Question]:
    - **IF [Observation] contains Multiple Images**:  The [Question] string MUST
    include at least 2 explicit image references (e.g., [Image 1, Image 2, Image3,
    ...]).
    - **IF [Observation] contains Only One Image**: You MUST generate a question
    specific to that image. Do NOT hallucinate other images. The [Question] string
    MUST include explicit image references (e.g., Image 1).
4. Fact-Based: The [Answer] must rely on the [Observation] text.
5. Atomic [Question]: the [Question] string must be a single query. It MUST NOT be
    a compound question or contain any sub-questions.

## OUTPUT FORMAT AND CONSTRAINTS (MUST FOLLOW):
Return a valid JSON **List** of objects:
```

---

```json
[
  {{
    "qa_pairs" : {{
      "qa1": {{
        "question": "...",
        "answer": "...",
      }},
      "qa2": {{
        "question": "...",
        "answer": "...",
      }},
      ...
    }},
    "image_indices": [...],
    "biomedical_entities": ["entity1", "entity2", "..."]
  }}
]
```

```
Task
Generate multiple visual description QA pairs based on the [Observation].
2. Extract a list of all unique biomedical entities (e.g., cell types, staining,
   anatomical structures) mentioned in the [Observation] and put them in
   "biomedical_entities".
[Your Output]:
```

### A.4.2. LOGIC CHAIN CONSTRUCTION

Constructs the structured reasoning path from experiment to conclusion.

---

**Logic Chain Generation Prompt**

```
You are a rigorous biomedical expert. You need to construct logical reasoning
    chains based on the provided Original Text and Visual Evidence.

Original Text:
{context}

Visual Evidence:
{observation}

Please integrate the Context and Visual Evidence to form detailed logical reasoning
    chains that lead to conclusions.
Requirements:
- Each independent research in the Context should correspond to a separate logical
    reasoning chain.
- Each logical reasoning chain should follow the logic:
  Research Context -> Experiments -> Conclusion
- Each Experiment should follow the logic:
  Experimental Setting -> Experiment Goal -> Visual Phenomenon -> Interpretation ->
    Sub-Conclusion
    - Experimental Setting: Describe the experimental setup, including materials,
    methods, and conditions.
    - Experiment Goal: Purpose of the experiment.
    - Visual Phenomenon: Specific **visual** observations from the experiment, not
    interpretations.
    - Interpretation: Scientific explanation of the visual phenomenon.
    - Sub-Conclusion: Conclusion drawn from the interpretation, related to the
    final conclusion.
```

---

```
- If a visual phenomenon of an experiment is mentioned in the Visual Evidence, mark
    which image it corresponds to in the format [Image X].
- Some experiments may not have any visual phenomenon in the Visual Evidence. There
    are two cases:
    - The Context provides the visual phenomenon directly. In this case, provide
    the visual phenomenon and mark it as [Context].
    - The visual phenomenon is missing. In this case, provide [Missing] as the
    visual phenomenon.
- Avoid precise numerical measurement data unless exactly the same numbers are
    present in Experimental Setting.
- The process of achieving the conclusion should include all necessary intermediate
    sub-conclusions and corresponding experiments.
- Each logical reasoning chain should end with a clear conclusion.
- If a certain experiment has no contribution to the final conclusion, it should be
    omitted from the whole logical reasoning chain.
Output Format:
Provide the logical reasoning chains in JSON format as a list of objects with the
    following structure:
```json
[
  {{
    "research_context": "Description of the research context.",
    "experiments": [
      {{
        "experimental_setting": "Description of the experimental setting.",
        "experiment_goal": "Description of the experiment goal.",
        "visual_phenomenon": "Visual phenomenon details with [Image X] or [Context]
    or [Missing].",
        "interpretation": "Interpretation of the visual phenomenon.",
        "sub_conclusion": "Conclusion drawn from the interpretation, related to the
    final conclusion."
      }},
      ...
    ],
    "reasoning": {{
      "intermediate_inferences": [
        {{
          "sub_conclusion": "Description of the intermediate inference.",
          "based_on_experiments": [Indices of experiments contributing to this
    inference]
        }},
        ...
      ],
      "content": "Detailed reasoning process leading to the conclusion.",
      "conclusion": "Final conclusion derived from the reasoning."
    }}
  }}
]
```
```

### A.4.3. OPEN-ENDED EXAM QUESTION GENERATION

Generates complex, open-ended questions based on the logic chain.

**Open-Ended Question Generation Prompt**

```
You are a biomedical expert.
You are given a logic chain, and you need to generate an exam question to test
    students' comprehension of the logic chain.
```

```
Logic Chain:
{logic_chain}

Some extra information that may help you:
Visual Evidence:
{visual_evidence}

Original Text:
{original_text}

The questions is expected to be hard, requiring both accurate observations and deep
    understanding of the logic chain. This includes the following aspects:
- For the answer:
    - The question should be open-ended.
    - The answer should contain the whole logical reasoning chain above.

- For the information provided in the question:
    - The Research Context should be provided.
    - The Setting of each Experiment should be provided.
    - The Goal of each Experiment should NEVER be provided.
    - For Visual Phenomenon of each Experiment:
       - If at least one visual phenomenon of the experiment is mentioned in the
    Visual Evidence, do NOT provide the Visual Phenomenon or Result. I.e. sentences
    like "[Image X] shows ..." should NEVER appear in the question.
          - Only one EXEMPT: If Visual Phenomenon contains Scale Bars, mention the
    scale ratio in the question.
       - If all visual phenomena of the experiment are provided in Context,
    provide the Visual Phenomenon. Do NOT provide the Result.
       - If the visual phenomenon is marked as Missing, provide the experiment
    result instead.
    - The direct result (NOT further Interpretation or Sub-conclusion) of each
    experiment should be provided only if the Visual Phenomenon is marked as
    Missing.
    - The Interpretation and Sub-Conclusion of each Experiment should NEVER be
    provided.
    - Intermediate Inferences should NEVER be provided.
    - Reasoning from Intermediate Inferences to Conclusion should NEVER be provided.
    - The Conclusion should NEVER be provided.

- For how to ask the question:
    - The question should not easily guide students to the answer. That is:
       - The question should not give any clues about how to reason to the answer.
       - The question should not give away intermediate steps or conclusions.

For example:
The logic chain is:

Research Context RC
Experiment E1:
  Setting: S1
  Visual Phenomenon: P1 [Image X]
  Interpretation: I1
  Sub-Conclusion: SC1
Experiment E2:
  Setting: S2
  Visual Phenomenon: P2 [Context]
  Interpretation: I2
  Sub-Conclusion: SC2
Experiment E3:
  Setting: S3
  Visual Phenomenon: [Missing]
  Interpretation: I3
  Sub-Conclusion: SC3
```

```
Final Reasoning:
  Based on SC1, SC2 and SC3, we conclude Conclusion C.

Do NOT ask:
- [BAD CASE] "How is conclusion C derived?" (gives away the conclusion)
- [BAD CASE] "Research RC, conducted experiment E1, setting S1, observed P1, ..."
    (gives away visible phenomenon in the images)
- [BAD CASE] "Research RC, ..., conducted experiment E2, setting S2, result R2,
    ..." (gives away experiment result where phenomenon is provided in Context)
- [BAD CASE] "Research RC, ..., conducted experiment E3, interpretation I3, ..."
    (gives away interpretation)
- [BAD CASE] "Research RC, conducted experiment E1, (Did not provide S1), ..."
    (misses the setup of an experiment)
- [BAD CASE] "Research RC, ..., conducted experiment E2, setting S2, (Did not
    provide P2), ..." (misses the visual phenomenon that is not provided in the
    images but provided in Context)
- [BAD CASE] "Research RC, ..., conducted experiment E3, setting S3, (Neither
    phenomenon nor direct result), ..." (misses the direct result of an experiment
    where the visual phenomenon is marked as Missing)
- [BAD CASE] "Based on SC1, SC2 and SC3, what is the conclusion?" (gives away
    intermediate inferences)
Ask instead:
[GOOD CASE] "Research RC, conducted experiment E1, setting S1; conducted experiment
    E2, setting S2, observed P2; conducted experiment E3, setting S3, got result
    R3. What can be concluded from these experiments?"
Note that you do not need to explicitly ask "Please give a detailed reasoning
    process". A clever student should know to provide the reasoning process to
    reach the conclusion.

Format your output as a JSON object with three fields: "question" and "answer",
    where "question" contains the generated question, "answer" and "explanation",
    where "explanation" explains how you generated this question-answer pair
    according to the requirements above.
```json
{{
  "explanation": "{{your explanation here}}",
  "question": "{{your question here}}",
  "answer": "{{your answer here}}"
}}
```
```

We employ a three-stage QC mechanism to ensure data integrity.

### A.4.4. QC STAGE 1: LOGIC INTEGRITY

Evaluates the internal coherence of the generated logic chain.

**QC Step 1 (Logic Check) Prompt**

```
You are an expert in biomedical reasoning and logic evaluation.
Your task is to evaluate the integrity and coherence of a logic chain.
The input is a structured list of strings representing the progression from
    experimental facts to intermediate inferences, and finally to a conclusion.

# Input Data

[Logic_Chain] (The reasoning path to evaluate):
{flattened_logic_chain}
```

```
# Evaluation Criteria (1-5 Scale)

1. Evidence Support Strength
   Assess if the intermediate inferences provide sufficient and accurate support
    for the final reasoning content.
    - Score 1 (Critical Fail): Contradictory or Unsupported. The final content makes
    claims that contradict the intermediate inferences or relies on evidence not
    present in the chain.
    - Score 3 (Borderline): Weak or Partial Support. The final content is somewhat
    related but contains major leaps in logic or includes details not fully backed
    by the intermediate steps.
    - Score 5 (Pass): Strong Support. The final content is a robust and accurate
    synthesis strictly derived from the provided intermediate inferences.

2. Logical Flow and Coherence
   Assess if the transition from Intermediate Inferences to the Final Conclusion is
    logically sound and seamless.
    - Score 1 (Critical Fail): Fragmented or Disjointed. The logic jumps randomly;
    the connection between the inference layer and the conclusion layer is broken
    or nonsensical.
    - Score 3 (Borderline): Rough or Repetitive. The flow is understandable but
    clunky, redundant, or requires the reader to guess the connection between steps.
    - Score 5 (Pass): Seamless and Coherent. The reasoning flows naturally like a
    scientific argument; the conclusion feels like the inevitable result of the
    preceding steps.

# Output Format (Strict JSON)

You must return the result strictly in the following format:

<scores>
{{
  "Evidence Support Strength": A,
  "Logical Flow and Coherence": B
}}
</scores>

<explanation>
[Provide a brief explanation for your scoring. explicitly stating if there are
    logical gaps, contradictions, or if the chain is solid.]
</explanation>

(Where A, B are integer scores from 1 to 5)
```

### A.4.5. QC STAGE 2: FACT GROUNDING

Verifies that visual descriptions in the logic chain are supported by the source observation.

```
QC Step 2 (Grounding Check) Prompt

You are an expert in biomedical text verification and fact-checking.
Your task is to verify if the [Visual Phenomena] described in the logic chain are
    supported by the provided Source Data ([Observation] and [Context]).

# Input Data

[Observation] (Objective visual descriptions of the images):
{Observation}

[Context] (Background containing [Image] tags):
```

```
{Context}

[Visual_Phenomena] (The descriptions extracted from the logic chain to be verified):
{VisualPhenomena}

# Evaluation Criteria (1-5 Scale)

1. Source Grounding & Verification
   Assess if every visual phenomenon listed in the Target is explicitly mentioned
    or clearly visible in the [Context] or [Observation].
    - Score 1 (Critical Fail): Hallucination. The target describes features that are
    completely absent from both the Observation and Context, or contradicts them.
    - Score 3 (Borderline): Partial Match. Some descriptions are supported, but
    others are missing source evidence, or the target adds significant details not
    found in the source.
    - Score 5 (Pass): Fully Grounded. Every statement in the [Visual Phenomena] is
    directly supported by evidence found in the Source Observation or Source
    Context (textual descriptions of visual outcomes).

# Output Format (Strict JSON)

You must return the result strictly in the following format:

<scores>
{{
  "Source Grounding & Verification": A
}}
</scores>

<explanation>
[Provide a brief explanation. If there is a hallucination or missing reference,
    explicitly quote the unsupported part.]
</explanation>

(Where A is an integer score from 1 to 5)
```

### A.4.6. QC STAGE 3: QUESTION-ANSWER ALIGNMENT

Ensures the final conclusion accurately answers the generated question.

**QC Step 3 (Alignment Check) Prompt**

```
You are an expert in evaluating question-answering logic.
Your task is to verify if the provided Conclusion effectively answers or
    corresponds to the specific Question asked.

# Input Data

Question:
{Question}

Observation (Visual Evidence containing scale info):
{Observation}

Logic Chain:
{logic_chain}

Conclusion (Derived from Logic Chain):
{Conclusion}
```

```
# Evaluation Criteria (1-5 Scale)

1. Question-Conclusion Alignment
   Assess if the Conclusion directly addresses the core inquiry of the Question.
    - Score 1 (Fail): The conclusion is irrelevant, unrelated, or contradicts the
    premise of the question. It does not provide an answer.
    - Score 3 (Passable): The conclusion is related and provides a partial answer,
    but may be slightly tangential or misses the specific format requested.
    - Score 5 (Pass): The conclusion provides a clear, logical, and direct answer to
    the question. It functions effectively as the final output.

2. Scale/Legend Consistency Check
Check if the problem statement lacks a scale/legend, but the observation results,
    reasoning content, and conclusion clearly include scale numbers or scale
    information.
- Score 1 point (Serious Failure): The problem statement lacks a scale/legend, but
    the observation results contain explicit scale numbers (e.g., "50 nm,"
    "scale"), and the reasoning content utilizes this scale information from the
    observation.
- Score 5 points (Pass): The problem statement and observation results are
    consistent; either both include a scale/legend, or neither includes
    scale-related information. If the problem statement includes scale-related
    information, but the conclusion and reasoning content do not use it, it is not
    considered an error.

3. Reasoning Validity
   Assess if the Logic Chain steps contain excessive speculation or hallucinations
    not supported by the Observation.
    - Score 1 (Critical Fail): Given ONLY Research Context, Experimental Settings,
    and Visual Phenomenon, The "inference", "sub_conclusion", "content", and
    "conclusion" parts contain details impossible to know.
    - Score 5 (Pass): Given ONLY Research Context, Experimental Settings, and Visual
    Phenomenon, the "inference", "sub_conclusion", "content", "conclusion" parts
    are all supported without any hallucination.

# Output Format (Strict JSON)

You must return the result strictly in the following format:

<scores>
{{
  "Question-Conclusion Alignment": A,
  "Scale/Legend Consistency Check" : B,
  "Reasoning Validity" : C
}}
</scores>

<explanation>
[Briefly explain why the conclusion satisfies or fails to answer the question.]
</explanation>

(Where A, B, and C are integer score from 1 to 5)
```

## B. QA Exmaple

As illustrated in Figure A1, each benchmark instance consists of (i) multimodal scientific evidence, including histological or immunohistochemical images together with experimental settings and background descriptions, (ii) a research-oriented question that explicitly targets biological mechanisms or functional interpretations, and (iii) a ground-truth reference that follows a structured reasoning chain: observations, interpretation, sub-conclusions, and final conclusion.

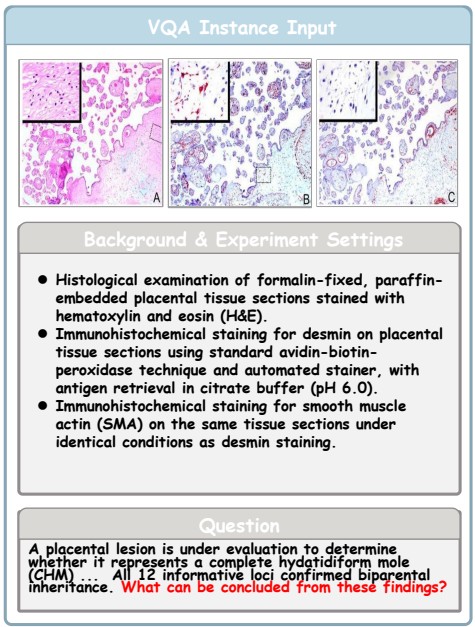 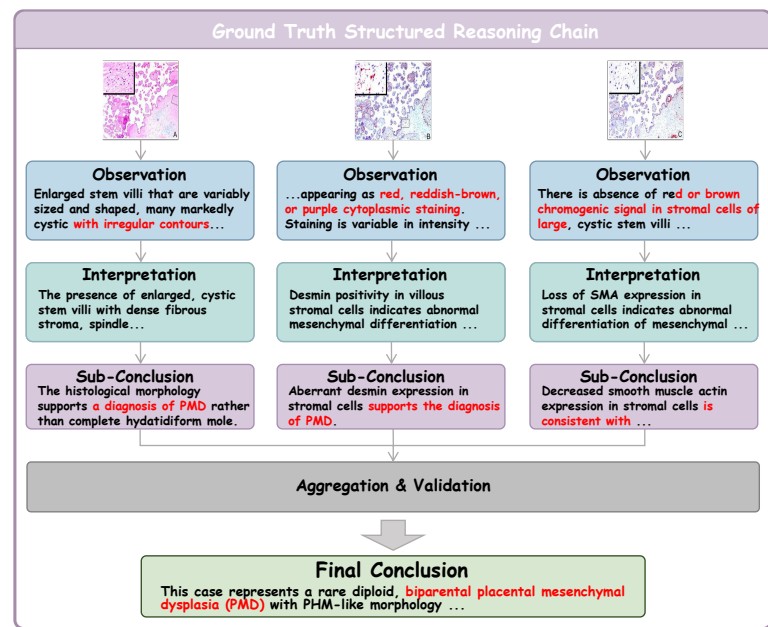

*Figure A1.* **Overview of a Representative SORBE Instance.** The left panel displays the multimodal input (visual evidence, experimental context, and query), while the right panel delineates the structured ground-truth reasoning chain, explicitly mapping visual observations to the final diagnostic conclusion via intermediate logical steps.

## C. Capability Coverage of SORBE

Figure A2 illustrates how SORBE evaluates scientific reasoning capabilities through representative multi-image biomedical questions. The left panel presents four example questions constructed from clusters of images originating from different experimental conditions and protocols, with annotated textual and visual components indicating how specific parts of each question probe distinct reasoning processes. The right panel of the figure summarizes a subset of the scientific reasoning capabilities systematically evaluated within the SORBE dataset, serving as illustrative examples of the broader capability space covered by the benchmark. Individual questions may involve multiple reasoning dimensions simultaneously, reflecting the multi-step, evidence-driven nature of real biomedical scientific reasoning. Together, these examples demonstrate that SORBE is designed to probe not isolated visual recognition, but structured reasoning across experimental context, visual evidence, and causal interpretation.

## D. Fine-grained Analysis of Representative Models

We conducted a detailed fine-grained analysis of three representative models—GPT-4o, Lingshu-32B, and Hulu-32B—to examine the relationship between outcome-level correctness and process-level reasoning quality on SORBE. The results of this analysis are presented in Figure A3. As shown in Figure A3(A), conclusion scores for all three models are largely concentrated in the 0.5–0.75 range, indicating that models frequently produce answers that appear moderately plausible at the outcome level. In contrast, Figure A3(B) shows that process scores are predominantly concentrated in the lowest range (0–0.25), suggesting that many intermediate reasoning steps exhibit limited alignment with the available experimental evidence. Figure A3(C) further categorizes process-level failures into three major types—visual errors or missing evidence, incorrect interpretation of observations, and difficulties in deriving evidence-supported conclusions—indicating that discrepancies arise at multiple stages of the reasoning process. Figures A3(D1–D3) illustrate a consistent upward

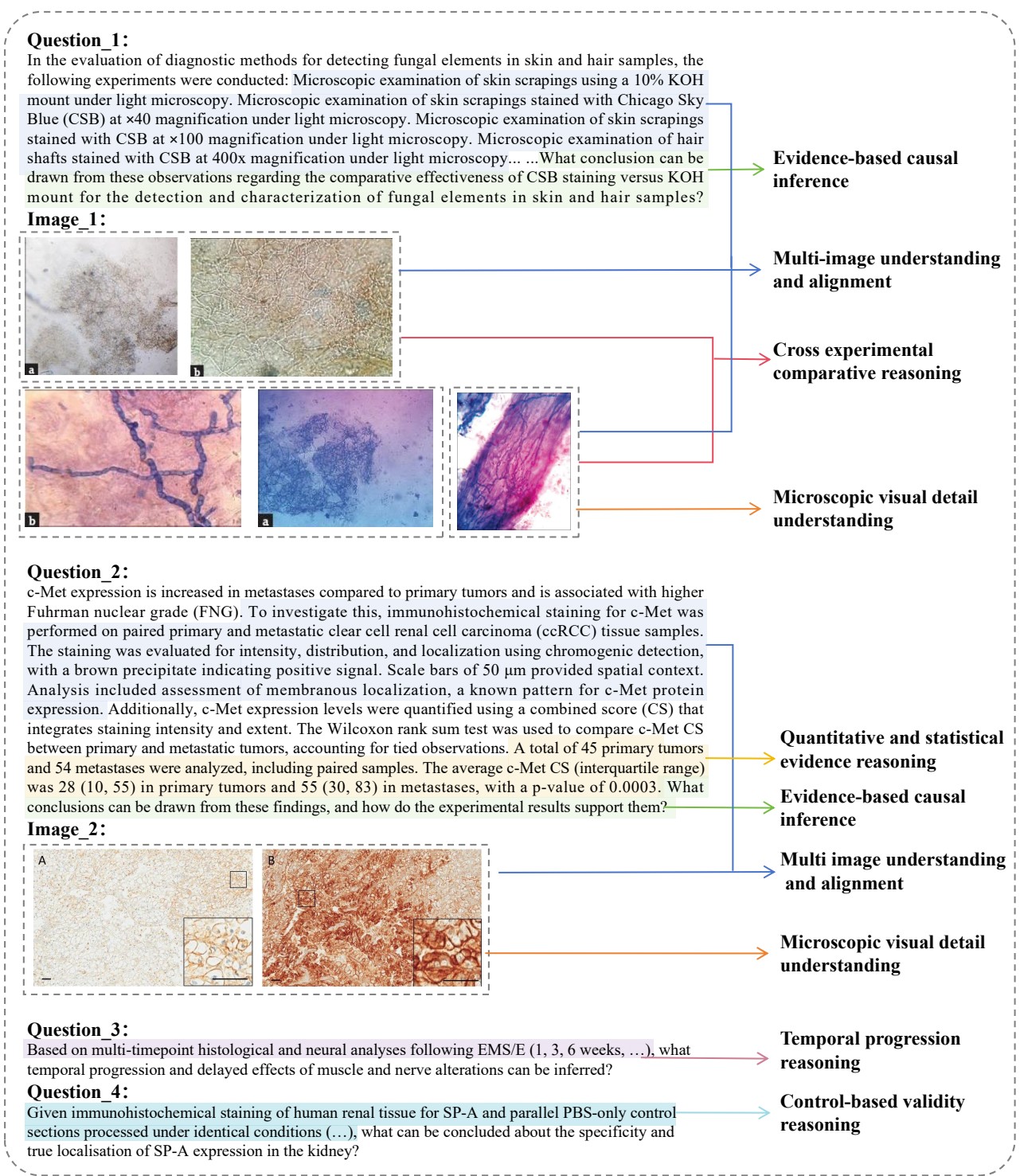

*Figure A2.* **Capability coverage of the SORBE benchmark.** Example questions on the left demonstrate how SORBE probes different classes of biomedical reasoning through annotated question components and visual evidence, while the right panel summarizes the scientific reasoning abilities systematically evaluated across the dataset.

trend between conclusion scores and average process scores across models; however, the corresponding scatter plots in Figures A3(D4–D6) reveal substantial variance, with process scores spanning a wide range even for high conclusion

scores. This pattern suggests that higher-quality conclusions do not necessarily correspond to consistently well-grounded reasoning processes. Collectively, these results highlight a notable divergence between outcome-oriented evaluation and process-oriented assessment, underscoring the importance of explicitly evaluating evidence alignment when assessing scientific reasoning in biomedical VQA.

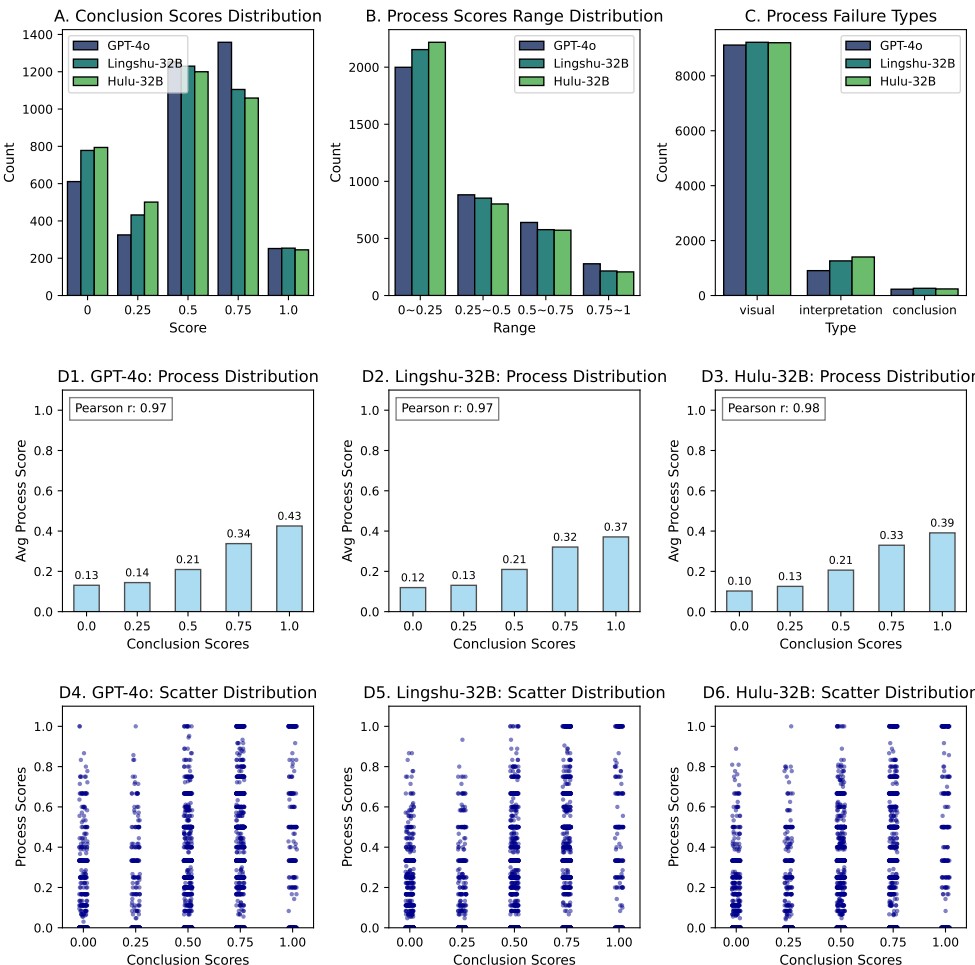

*Figure A3.* Detailed analysis of three representative models (GPT-4o, Lingshu-32B, Hulu-32B) on SORBE. (A) Conclusion score distribution; (B) Process score range distribution; (C) Process failure types; (D1-D3) Relationship between conclusion scores and average process scores for each model; (D4-D6) Scatter distribution of conclusion scores versus process scores.

## E. Ablation study of different model evaluation

As shown in Tab 6, this study evaluates three medical language models using two distinct evaluators: Qwen3-235B-A22B-Instruct-2507 and DeepSeek-V3.2. Both evaluators consistently rank GPT-4o as the top performer across all three metrics, with Lingshu-32B and Hulu-Med-32B following in that order. This consistency suggests robust performance differences between models regardless of the evaluation framework. Notably, the performance gaps are more pronounced in DeepSeek evaluations (0.4–3.8 percentage points between models) compared to Qwen3 evaluations (0.9–2.1 percentage points).

*Table 6.* Ablation study results from different model evaluations.

| Qwen3 Evaluation | | | |
| --- | --- | --- | --- |
| Model | LCRS (%) | PROC (%) | CONC (%) |
| GPT-4o | 34.1 | 33.1 | 60.9 |
| Lingshu-32B | 32.9 | 32.9 | 57.3 |
| Hulu-Med-32B | 32.0 | 31.6 | 56.9 |
| DeepSeek Evaluation | | | |
| Model | LCRS (%) | PROC (%) | CONC (%) |
| GPT-4o | 26.0 | 25.8 | 52.3 |
| Lingshu-32B | 22.6 | 23.1 | 47.7 |
| Hulu-Med-32B | 22.2 | 22.4 | 46.6 |

# F. Error Analysis

In this section, we present a detailed qualitative error analysis of four representative cases generated by **GPT-4o**, **Gemini-2.5-Flash**, **Lingshu-32B**, and **Qwen3-VL-235B** on the SORBE benchmark. We specifically isolate instances exhibiting *process-outcome misalignment*, where models reach the correct final conclusion ($S_{conc} = 1$) despite relying on flawed, hallucinated reasoning paths ($S_{proc} = 0$). These examples empirically demonstrate the phenomenon of "semantic plausibility" and underscore the critical necessity of the Logic-Coupled Reasoning (LCR) metric in preventing the overestimation of model capabilities in rigorous biomedical contexts.

## Gemini-2.5-Flash

**Visual Evidence**

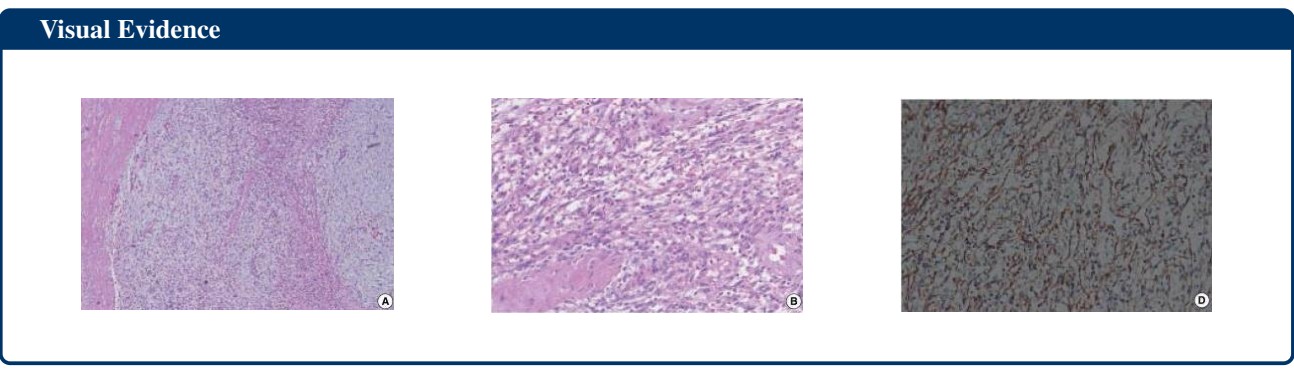

**Question**

The case report aims to confirm the diagnosis of a renal mass as inflammatory myofibroblastic tumor (IMT) based on histopathological and immunohistochemical findings, distinguishing it from renal cell carcinoma and other malignant or benign renal tumors. To evaluate the architectural and cytological features of the tumor, a histopathological examination of the resected renal tumor was performed using hematoxylin and eosin (H&E) staining, viewed under light microscopy at low and high magnification. To assess cellular morphology and architectural patterns in greater detail, high-magnification H&E staining was used. Additionally, immunohistochemical staining was performed using antibodies against epithelial membrane antigen (EMA), actin, and other markers, with visualization via chromogenic detection. What can be concluded from these experiments regarding the nature and diagnosis of the renal mass?

# Logic Chain Construction

> **Experiment 1**
>
> **Visual Phenomenon:**
> A thick, pink-stained, eosinophilic collagenous fibrous capsule is visible on the left side of the image, adjacent to a myxoid zone with a pale, bluish, gelatinous matrix containing scattered inflammatory cells. Centrally, there is a cellular region with spindle-shaped cells arranged in interlacing fascicles [Image 1].
>
> **Interpretation:**
> The eosinophilic capsule indicates a well-encapsulated tumor. The myxoid stroma with inflammatory infiltrate and the fascicular arrangement of spindle cells are characteristic of IMT, which typically shows a triphasic histologic pattern including fibrous, myxoid, and cellular zones.
>
> *Sub-Conclusion:*
> *The tumor exhibits histological features consistent with a well-encapsulated mesenchymal neoplasm showing myxoid and cellular zones with inflammatory infiltrate, supporting a diagnosis of IMT.*

**Experiment 2**

**Visual Phenomenon:**

Densely packed spindle-shaped cells arranged in parallel and interlacing fascicles with occasional whorled patterns. Cells have elongated, hyperchromatic nuclei, high nuclear-to-cytoplasmic ratio, minimal extracellular matrix, and occasional mitotic figures. No necrosis or epithelial structures are present [Image 2].

**Interpretation:**

The fascicular and whorled growth of uniform spindle cells with minimal atypia and low mitotic rate, in the absence of necrosis or epithelial differentiation, is typical of IMT. This contrasts with high-grade sarcomas or sarcomatoid carcinoma, which usually show marked atypia, frequent mitoses, and necrosis.

*Sub-Conclusion:*
*The absence of high-grade malignant features and the presence of a fascicular spindle cell pattern support a benign or low-grade neoplastic process such as IMT rather than a high-grade sarcoma or renal cell carcinoma.*

**Experiment 3**

**Visual Phenomenon:**

Brownish cytoplasmic staining of variable intensity in spindle-shaped cells arranged in interlacing fascicles and whorls. Staining is heterogeneous, with clusters of strongly positive cells and others showing weak or no staining. No glandular structures are present [Image 3].

**Interpretation:**

The brown cytoplasmic staining indicates positive immunoreactivity for a marker expressed in myofibroblasts. Given the context, this corresponds to focal positivity for actin and EMA, markers of myofibroblastic differentiation. Vimentin positivity (diffuse) further supports mesenchymal origin. Absence of cytokeratin, desmin, CD34, etc., rules out carcinomas, rhabdomyosarcoma, and vascular tumors.

*Sub-Conclusion:*
*The immunohistochemical profile—vimentin diffuse (+), actin and EMA focal (+), and negative for epithelial, melanocytic, and vascular markers—confirms the myofibroblastic nature of the tumor and supports the diagnosis of IMT.*

**Final Conclusion (Reasoning Synthesis)**

The renal mass is definitively diagnosed as inflammatory myofibroblastic tumor (IMT) based on its encapsulated architecture, spindle cell morphology with inflammatory background, and immunohistochemical profile confirming myofibroblastic origin, despite initial clinical suspicion of renal cell carcinoma.

## Evaluation Comparison

### Model Answer

Based on the provided histopathological and immunohistochemical findings, the following conclusions can be drawn regarding the nature and diagnosis of the renal mass: 1. Histopathological Features (H&E Staining - Images A and B): Low Magnification (Image A): This image reveals a cellular tumor composed predominantly of spindle-shaped cells, with an admixture of inflammatory cells. The tumor appears relatively well-circumscribed from the surrounding renal parenchyma on the left. The overall architecture is infiltrative with a somewhat uniform, cellular pattern. This appearance is atypical for conventional renal cell carcinoma, which often presents with clear cells, granular cells, or papillary/tubular structures. High Magnification (Image B): This image provides a more detailed view of the cellular morphology. It clearly shows a proliferation of elongated, spindle-shaped cells arranged in fascicles or haphazardly. Crucially, there is a prominent admixed inflammatory infiltrate, including lymphocytes and plasma cells (visible as small, dark nuclei), which is a characteristic feature of inflammatory myofibroblastic tumors. While mitotic figures are not overtly prominent in this specific field, the spindle cell morphology with an inflammatory background is highly suggestive of IMT. The absence of significant cellular atypia, pleomorphism, or overt sarcomatoid features often seen in high-grade sarcomas or sarcomatoid renal cell carcinoma also leans towards IMT or a low-grade mesenchymal lesion. Some larger, multinucleated cells (possibly giant cells) can also be seen, which are sometimes present in IMT.2. Immunohistochemical Features (Actin Staining - Image D): Image D demonstrates strong brown staining within the spindle cells, indicating positive expression for actin. This positivity for actin (typically smooth muscle actin, SMA) is a key immunohistochemical marker for myofibroblastic differentiation. Inflammatory myofibroblastic tumors are characterized by the proliferation of myofibroblasts, which express SMA. The diffuse and strong positivity in the spindle cells confirms their myofibroblastic nature. The question also mentions that EMA (epithelial membrane antigen) was used. If this were an IMT, EMA would typically be negative, which would help rule out epithelial tumors like renal cell carcinoma. Renal cell carcinomas are generally positive for epithelial markers (like EMA and cytokeratins).Conclusion: Combining the histopathological and immunohistochemical findings, the renal mass exhibits: A proliferation of spindle cells with an admixed prominent inflammatory infiltrate (lymphocytes and plasma cells) on H&E staining. Strong positivity for actin (indicating myofibroblastic differentiation) in the spindle cells.These features are classic and highly characteristic of an Inflammatory Myofibroblastic Tumor (IMT). The findings effectively differentiate it from typical renal cell carcinoma (which would be epithelial, EMA/cytokeratin positive, and usually SMA negative), as well as many other benign or malignant renal tumors that would have different cellular compositions, architectural patterns, and immunohistochemical profiles. Therefore, these experiments strongly support the diagnosis of inflammatory myofibroblastic tumor for the renal mass.

The histopathological examination using low-magnification H&E staining reveals a well-encapsulated renal mass with a triphasic architectural pattern: an outer dense eosinophilic collagenous capsule, an adjacent myxoid zone with a pale, bluish, gelatinous matrix containing scattered inflammatory cells, and a central cellular region composed of spindle-shaped cells arranged in interlacing fascicles. This zonation—fibrous, myxoid, and cellular—is characteristic of inflammatory myofibroblastic tumor (IMT), as described by Coffin et al., and supports a mesenchymal neoplasm with myofibroblastic and inflammatory components. High-magnification H&E staining shows densely packed spindle-shaped cells arranged in parallel and interlacing fascicles with occasional whorled patterns. The cells exhibit elongated, hyperchromatic nuclei, a high nuclear-to-cytoplasmic ratio, minimal extracellular matrix, and occasional mitotic figures, but no tumor necrosis or epithelial structures are present. The absence of significant nuclear atypia, frequent mitoses, or necrosis argues against high-grade sarcomas or sarcomatoid renal cell carcinoma, which typically display marked pleomorphism and necrotic debris. The uniformity of the spindle cells and their fascicular growth pattern are consistent with a low-grade or benign mesenchymal neoplasm rather than a malignant epithelial or sarcomatous process. Immunohistochemical staining demonstrates brown cytoplasmic reactivity in the spindle cells, with variable intensity and heterogeneous distribution—some areas show strong staining while others are weak or negative. The positive immunoreactivity corresponds to focal expression of epithelial membrane antigen (EMA) and actin, both markers associated with myofibroblastic differentiation. Vimentin is diffusely positive, confirming mesenchymal origin. Critically, the tumor cells are negative for cytokeratin (ruling out carcinoma), desmin (ruling out rhabdomyosarcoma), CD34 (excluding vascular tumors), HMB45 (excluding melanocytic lesions), S-100, and other lineage-specific markers, effectively eliminating other spindle cell neoplasms and epithelial malignancies such as renal cell carcinoma or angiomyolipoma. Together, these findings confirm that the tumor is a well-encapsulated mesenchymal neoplasm composed of spindle cells with myofibroblastic differentiation, embedded in a variable stroma with inflammatory infiltrate, and lacking features of high-grade malignancy or epithelial lineage. The histologic architecture, cytological features, and immunophenotypic profile collectively support a definitive diagnosis of inflammatory myofibroblastic tumor (IMT), a lesion that clinically mimics renal cell carcinoma but is distinct in its pathogenesis and behavior.

# GPT-4o

## Visual Evidence

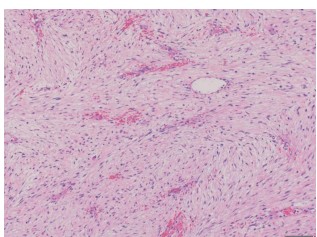 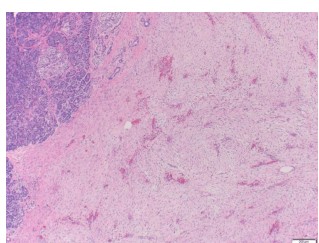 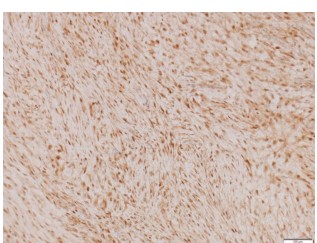

## Question

A 39-year-old woman presented with fever, chills, severe abdominal pain, and peritonitis. Imaging revealed an abdominal mass with possible free air, and exploratory laparotomy showed a perforated mass with abscess formation. She underwent en-bloc resection of the mass involving the omentum, stomach, colon, and pancreas. To determine the histopathological diagnosis, the following studies were performed: - Histological examination of the resected mass was conducted using hematoxylin and eosin (H&E) staining at 100x magnification, as indicated by a 100-micrometer scale bar. - Additional histological evaluation was performed on a tissue section containing both pancreatic parenchyma and adjacent fibrous tissue, using H&E staining at 40x magnification. - Immunohistochemical staining was carried out on a tissue section using a chromogen to detect protein expression, visualized at 100x magnification with a 100-micrometer scale bar. What can be concluded from these studies regarding the nature and diagnosis of the abdominal mass?

# Logic Chain Construction

### Experiment 1

**Visual Phenomenon:**
The tissue consists of sweeping, interlacing fascicles of spindled cells with uniform elongated nuclei, minimal pleomorphism, finely dispersed chromatin, and inconspicuous nucleoli. Cytoplasmic borders are poorly defined, giving a syncytial appearance. The stroma is pale eosinophilic and collagenous. Slit-like vascular channels are present, some containing erythrocytes, and scattered extracellular erythrocytes are seen. The growth pattern is disorganized and infiltrative without encapsulation [Image 1].

**Interpretation:**
The uniform spindle cells with low-grade nuclear features and infiltrative growth are characteristic of a locally aggressive fibroblastic/myofibroblastic neoplasm. The absence of significant atypia, mitotic activity, or necrosis argues against sarcoma. The infiltrative, non-encapsulated pattern suggests locally invasive behavior.

*Sub-Conclusion:*
*The histological features are consistent with a benign but locally infiltrative mesenchymal neoplasm, specifically favoring desmoid-type fibromatosis.*

## Experiment 2

**Visual Phenomenon:**

On the left, organized pancreatic acinar and ductal structures are present. On the right, dense fibrous tissue composed of spindle-shaped cells in interlacing fascicles within a collagenous stroma is seen. The boundary is irregular, with spindle cell strands extending into the pancreatic parenchyma. The fibrous tissue has a pinkish hue with scattered blood vessels and inflammatory cells. No clear demarcation exists between the two tissues [Image 2].

**Interpretation:**

The irregular, interdigitating interface with tumor cells invading into normal pancreatic architecture demonstrates true infiltration, a hallmark of desmoid-type fibromatosis. The lack of a pushing border or capsule supports aggressive local behavior despite benign cytology.

*Sub-Conclusion:*

*The tumor exhibits infiltrative growth into adjacent pancreatic tissue, supporting a diagnosis of desmoid-type fibromatosis rather than a benign encapsulated fibrous lesion.*

## Experiment 3

**Visual Phenomenon:**

Brown chromogenic staining is localized to both nuclear and cytoplasmic compartments, with variable intensity and heterogeneous distribution. Nuclear staining is prominent, with elongated or ovoid nuclei aligned with the cell axis. Cytoplasmic staining is lighter and more diffuse. Cellular borders are indistinct [Image 3].

**Interpretation:**

Nuclear localization of brown chromogen is highly suggestive of nuclear $\beta$-catenin accumulation, which results from activating mutations in the WNT signaling pathway (e.g., CTNNB1). This is a molecular hallmark of desmoid-type fibromatosis. The heterogeneous staining pattern may reflect regional variation in protein expression or fixation.

*Sub-Conclusion:*

*Nuclear $\beta$-catenin immunoreactivity supports the diagnosis of desmoid-type fibromatosis.*

## Final Conclusion (Reasoning Synthesis)

The abdominal mass is definitively diagnosed as desmoid-type fibromatosis based on histomorphology, infiltrative growth pattern, and nuclear $\beta$-catenin expression.

## Evaluation Comparison

### Model Answer

Based on the provided information and images, here is a detailed analysis of the case: 1. Histological Examination of the Resected Mass (Image 1): - The image shows spindle-shaped cells forming fascicles, which is characteristic of a fibrous or smooth muscle origin. The background stroma is collagen-rich, suggesting a fibrous nature. This pattern is typical of a desmoid tumor or a fibrous proliferation. 2. Histological Evaluation of Tissue Section With Pancreatic Parenchyma (Image 2): - This image includes both pancreatic tissue and adjacent fibrous tissue. The pancreatic parenchyma appears normal, whereas the fibrous tissue shows a similar pattern to Image 1, indicating the mass is not originating from the pancreatic cells but rather from the adjacent fibrous tissue. 3. Immunohistochemical Staining (Image 3): - Immunohistochemistry results show uniform staining of the spindle cells, which suggests the expression of proteins characteristic of the mass's origin. The staining is consistent with markers that would typically highlight fibrous or fibroblastic tissue, such as vimentin. Conclusion: Considering the spindle cell morphology, fibrous tissue background, and positive immunohistochemical staining, the abdominal mass is best characterized as a desmoid-type fibromatosis. This is a locally aggressive fibroblastic proliferation that does not metastasize but can invade local structures, explaining the mass's involvement with multiple organs in the abdominal cavity. Final Diagnosis: Desmoid-type fibromatosis (aggressive fibromatosis). This diagnosis accounts for the histological and immunohistochemical findings, as well as the clinical presentation of a large, aggressive mass leading to the perforation and abscess formation.

### Ground Truth

The histological examination of the resected abdominal mass at 100x magnification reveals a proliferation of spindled cells arranged in sweeping, interlacing fascicles with uniform elongated nuclei, minimal nuclear pleomorphism, finely dispersed chromatin, and inconspicuous nucleoli. The cytoplasmic borders are poorly defined, giving the tissue a syncytial appearance. The stroma is pale eosinophilic and collagenous. Slit-like vascular channels are present, some containing erythrocytes, and scattered extracellular erythrocytes are observed. The growth pattern is disorganized and infiltrative, with no encapsulation. The evaluation of the tissue section containing both pancreatic parenchyma and adjacent fibrous tissue at 40x magnification shows organized pancreatic acinar and ductal structures on one side and dense fibrous tissue composed of spindle-shaped cells in interlacing fascicles on the other. The boundary between the two is irregular, with strands of spindle cells extending into the pancreatic parenchyma. There is no clear demarcation or pushing border, and the fibrous tissue contains scattered blood vessels and inflammatory cells. The immunohistochemical staining at 100x magnification demonstrates brown chromogenic signal localized to both the nucleus and cytoplasm of the spindle cells, with variable intensity and heterogeneous distribution. Nuclear staining is prominent, with elongated or ovoid nuclei aligned with the cell axis, while cytoplasmic staining is lighter and more diffuse. Cellular borders are indistinct. Together, these findings indicate a neoplasm composed of uniform spindle-shaped myofibroblast-like cells arranged in infiltrative fascicles within a collagen-rich stroma. The cytological features are bland, with no significant atypia, mitotic activity, or necrosis, ruling out a high-grade sarcoma. The infiltrative growth pattern, lack of encapsulation, and direct invasion into the pancreatic parenchyma demonstrate locally aggressive behavior despite benign cytology. This architectural pattern is characteristic of a locally invasive fibroblastic/myofibroblastic lesion. The immunohistochemical detection of nuclear $\beta$-catenin accumulation—evidenced by nuclear brown chromogen—indicates aberrant activation of the WNT signaling pathway, which is a molecular hallmark of desmoid-type fibromatosis, occurring in over 85% of sporadic cases. Correlating the histomorphology, infiltrative growth into adjacent organs, and nuclear $\beta$-catenin expression, the findings are diagnostic of desmoid-type fibromatosis. The extensive resection was necessary due to involvement of multiple abdominal structures, including the omentum, stomach, colon, and pancreas. The absence of tumor recurrence at 16 months post-resection and negative surgical margins support the diagnosis and indicate favorable local control, consistent with the clinical behavior of desmoid-type fibromatosis.

## Lingshu-32B

---

**Visual Evidence**

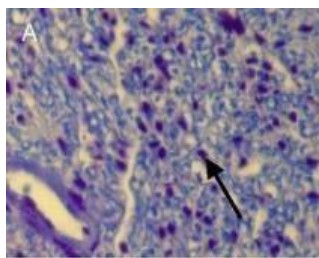 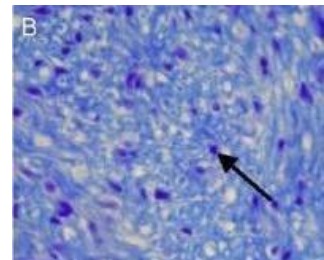 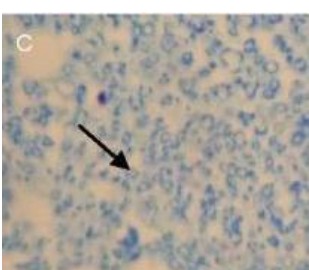

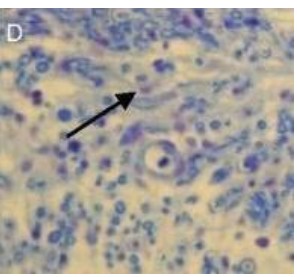

---

**Question**

A study investigated the effects of astragaloside IV on remyelination of the sciatic nerve in mice after injury, with assessments conducted at 8 weeks post-injury using Luxol fast blue staining to evaluate myelin sheath structure. Mouse sciatic nerve tissues were collected from animals treated with high-, moderate-, and low-dose astragaloside IV, as well as from untreated model mice. Histological sections were prepared and stained to visualize myelin architecture. In the high-dose and moderate-dose astragaloside IV groups, the tissues exhibited a distinct staining pattern with well-defined structural features. In the low-dose astragaloside IV group, the tissues showed a different staining pattern with additional morphological characteristics. The untreated model group displayed yet another distinct staining pattern, with notable structural abnormalities. A separate representative image confirmed the staining contrast between myelin and axonal structures across the tissue sections. Based on these experimental observations, what comprehensive conclusion can be drawn regarding the effects of astragaloside IV on sciatic nerve remyelination?

## Logic Chain Construction

### Experiment 1

**Visual Phenomenon:**
In the high-dose and moderate-dose astragaloside IV groups, the myelin sheaths appear regular, uniform, and densely packed, with a clear outline and consistent blue staining intensity surrounding lighter axonal profiles. Numerous dark purple nuclei are embedded in a pale matrix. A black arrow highlights a prominent region of compact myelin [Image 1].

**Interpretation:**
The regular, uniform, and densely packed blue-stained myelin sheaths indicate well-organized remyelination and preserved nerve architecture, suggesting effective regeneration of myelinated nerve fibers.

*Sub-Conclusion:*
*High- and moderate-dose astragaloside IV treatment is associated with structurally intact and well-organized myelin sheaths, indicative of successful remyelination.*

### Experiment 2

**Visual Phenomenon:**
The myelin sheaths show irregular arrangement with a clear outline, and fibrous connective tissue hyperplasia is visible among nerve bundles. Sky blue staining is present but with variable intensity. A black arrow points to a darkly stained ovoid structure within a loosely organized fibrous matrix [Image 2].

**Interpretation:**
Irregular myelin organization and fibrous hyperplasia suggest incomplete or disorganized regeneration, indicating suboptimal remyelination despite some myelin presence.

*Sub-Conclusion:*
*Low-dose astragaloside IV promotes limited remyelination with structural disorganization and fibrotic changes, indicating partial therapeutic efficacy.*

### Experiment 3

**Visual Phenomenon:**
The myelin sheaths are irregularly arranged, with fibrous connective tissue hyperplasia apparent. The black arrow indicates a region of irregular myelin. The staining shows heterogeneous blue intensity with granular and speckled distribution [Image 3].

**Interpretation:**
The irregular myelin architecture and fibrous hyperplasia reflect poor endogenous regeneration and scar tissue formation, consistent with incomplete nerve recovery in the absence of intervention.

*Sub-Conclusion:*
*The model group exhibits poor myelin regeneration and significant fibrotic changes, confirming impaired spontaneous remyelination after sciatic nerve injury.*

**Experiment 4**

**Visual Phenomenon:**

Blue-stained myelin sheaths appear as distinct circular and ovoid structures, with lighter unstained axonal profiles visible within. The background is pale, enhancing contrast. A black arrow highlights a specific myelin sheath [Image 4].

**Interpretation:**

The clear contrast between blue myelin and unstained axons allows for accurate morphological evaluation of myelinated fiber density and distribution.

*Sub-Conclusion:*

*Image 4 confirms the technical adequacy of Luxol fast blue staining for visualizing myelinated nerve fibers, supporting the validity of comparative assessments across groups.*

**Final Conclusion (Reasoning Synthesis)**

Astragaloside IV promotes dose-dependent remyelination of the sciatic nerve in mice after injury, with high and moderate doses significantly enhancing myelin sheath regeneration and structural organization compared to low dose and untreated controls, as evidenced by Luxol fast blue staining and histological analysis at 8 weeks post-injury.

**Evaluation Comparison**

Model Answer

To determine the effects of astragaloside IV on sciatic nerve remyelination, we analyze the histological images stained with Luxol fast blue, which highlights myelin sheaths. In the high-dose and moderate-dose astragaloside IV groups, the images show well-defined myelin structures, indicating effective remyelination. This suggests that higher doses of astragaloside IV promote better recovery of myelin integrity. In contrast, the low-dose group exhibits a different staining pattern with less defined myelin structures, implying partial remyelination but not as effective as higher doses. The untreated group shows significant structural abnormalities, with poorly defined myelin, indicating minimal or no remyelination. The additional representative image confirms the contrast between myelin and axonal structures, supporting the observed differences in remyelination across treatment groups. Therefore, the comprehensive conclusion is that astragaloside IV enhances remyelination in a dose-dependent manner, with higher doses leading to more effective restoration of myelin sheaths in the sciatic nerve.",

Ground Truth

The study evaluates the role of astragaloside IV in promoting remyelination of the sciatic nerve in mice following injury, using Luxol fast blue staining at 8 weeks post-injury to assess myelin sheath integrity and organization. In the high-dose and moderate-dose astragaloside IV groups, the myelin sheaths appear regular, uniform, and densely packed, with clear outlines and consistent blue staining intensity surrounding lighter axonal profiles. Numerous dark purple nuclei are embedded in a pale matrix, and a prominent region of compact myelin is highlighted, indicating well-organized and structurally preserved nerve architecture. This suggests effective regeneration of myelinated nerve fibers and robust remyelination in these groups. In contrast, the low-dose astragaloside IV group shows irregularly arranged myelin sheaths with variable staining intensity and visible fibrous connective tissue hyperplasia among nerve bundles. A darkly stained ovoid structure is observed within a loosely organized fibrous matrix, indicating incomplete or disorganized regeneration, with partial myelin formation but structural disarray and early fibrotic changes. This reflects suboptimal remyelination despite some therapeutic activity. The untreated model group exhibits the most severe deficits: myelin sheaths are irregularly arranged with heterogeneous, granular, and speckled blue staining, and fibrous connective tissue hyperplasia is prominent. A region of irregular myelin is indicated, consistent with poor endogenous regenerative capacity and scar tissue formation, confirming impaired spontaneous recovery after nerve injury. Finally, a representative image demonstrates clear contrast between blue-stained myelin sheaths and unstained axonal profiles against a pale background, with a specific myelin sheath highlighted, confirming the technical reliability of Luxol fast blue staining for morphological assessment of myelinated fibers. The consistent visualization across groups supports valid comparative analysis. Together, these observations reveal a dose-dependent effect of astragaloside IV on remyelination: high and moderate doses promote well-organized, dense, and structurally intact myelin regeneration, while low dose results in partial and disorganized remyelination with fibrosis. The untreated group shows minimal recovery with extensive fibrotic changes. Quantitative data referenced in Table 4 further support these findings, showing significantly higher numbers and diameters of myelinated fibers in all astragaloside IV-treated groups compared to the model group, with superior outcomes in high- and moderate-dose groups over low-dose, and no significant difference between high and moderate doses. Therefore, the evidence demonstrates that astragaloside IV promotes dose-dependent remyelination of the sciatic nerve in mice after injury, with high and moderate doses significantly enhancing myelin sheath regeneration and structural organization compared to low dose and untreated controls, as evidenced by histological analysis and Luxol fast blue staining at 8 weeks post-injury.

## Qwen3-VL-235B-A22B-Instruct

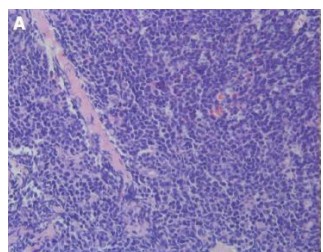
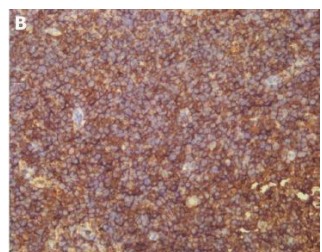
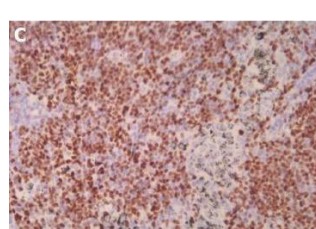

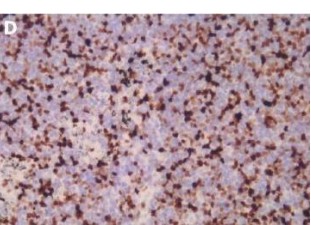

**Question**

A 65-year-old male presents with cough and dyspnea. Flexible bronchoscopy reveals mucosal infiltrative changes and bronchial obstruction at the right upper lobe inlet, while the remainder of the bronchial tree appears normal. To evaluate the nature of the lymphoid proliferation and confirm a diagnosis, the following investigations were performed: - A mucosal endobronchial biopsy from the right upper lobe inlet was examined using hematoxylin and eosin (H&E) staining. - Immunohistochemical staining for CD5 was performed on the same endobronchial biopsy. - Immunohistochemical staining for Cyclin D1, along with Bcl-2, CD19, CD20, and additional markers, was performed on the endobronchial biopsy and confirmed on a subsequent mediastinal lymph node biopsy obtained via mediastinoscopy. The cells in the endobronchial biopsy were positive for CD20, Bcl-2, CD5, CD19, and Cyclin D1. The mediastinal lymph node biopsy confirmed positivity for CD20, CD5, CD19, Bcl-2, Cyclin D1, and SOX11, and were negative for CD3, CD10, CD21, CD23, and Bcl-6. - Immunohistochemical staining for Ki67 was performed on the mucosal endobronchial biopsy. What can be concluded from these findings?

# Logic Chain Construction

## Experiment 1

**Visual Phenomenon:**
A densely cellular field with a diffuse, monotonous infiltrate of numerous small, round to oval cells with darkly stained, basophilic nuclei and condensed chromatin, some containing visible nucleoli. The cells exhibit a high nuclear-to-cytoplasmic ratio with scant, pale eosinophilic cytoplasm. Lymphoid follicles replace normal bronchial mucosal structures and are embedded in fibrous stroma. A thin, elongated, pale pink stromal or vascular element traverses the field diagonally. Occasional small eosinophilic granules or debris are visible in intercellular spaces [Image 1].

**Interpretation:**
The morphological features are consistent with a dense lymphoid infiltrate composed of small to medium-sized B-lymphocytes. The monotonous appearance, high nuclear-to-cytoplasmic ratio, and lack of glandular or epithelial organization suggest a neoplastic rather than reactive process. The presence of lymphoid follicles in fibrous stroma supports lymphoid hyperplasia, but the monomorphism raises suspicion for a low-grade lymphoma.

*Sub-Conclusion:*
*The histopathological features show a dense, monotonous lymphoid infiltrate suggestive of a B-cell lymphoproliferative disorder, possibly MCL.*

## Experiment 2

**Visual Phenomenon:**
A dense aggregation of small to medium-sized round cells with darkly stained nuclei and scant cytoplasm, arranged in a closely packed, diffuse, and confluent pattern. The majority of cells show a uniform brown chromogenic signal, indicating CD5 immunoreactivity. The background matrix is lightly stained and granular. Unstained areas correspond to stromal or vascular elements [Image 2].

**Interpretation:**
The uniform and intense brown chromogen deposition in the cytoplasm and/or membrane of the lymphoid cells indicates strong CD5 expression. CD5 positivity in a B-cell population is unusual and is typically seen in chronic lymphocytic leukemia (CLL) and mantle cell lymphoma (MCL), but not in most other B-cell lymphomas.

*Sub-Conclusion:*
*The lymphoid cells express CD5, narrowing the differential diagnosis to CD5-positive B-cell lymphomas such as MCL or CLL.*

## Experiment 3

**Visual Phenomenon:**

In the endobronchial biopsy, immunohistochemical staining reveals prominent brownish-red chromogen deposition predominantly within nuclei, with some cytoplasmic staining. Staining intensity is heterogeneous across the tissue. Lighter regions correspond to stromal or non-reactive areas [Image 3]. [Context] reports that the cells were positive for CD20, Bcl-2, CD5, CD19, and Cyclin D1 in the endobronchial biopsy, and the mediastinal lymph node biopsy confirmed positivity for CD20, CD5, CD19, Bcl-2, Cyclin D1, and SOX11, with negativity for CD3, CD10, CD21, CD23, and Bcl-6.

**Interpretation:**

Nuclear expression of Cyclin D1 is a hallmark of MCL, resulting from t(11;14) translocation. Combined with CD5 and CD20 positivity and absence of CD10 and CD23, this immunophenotype is highly specific for MCL. SOX11 positivity further supports classical MCL. The B-cell lineage (CD19, CD20) and anti-apoptotic marker Bcl-2 are consistent with MCL biology.

*Sub-Conclusion:*

*The immunophenotype (CD5+, CD20+, Cyclin D1+, SOX11+, CD10-, CD23-) is diagnostic of mantle cell lymphoma.*

## Experiment 4

**Visual Phenomenon:**

Brown nuclear labeling is observed in approximately 25% of the lymphoid cells, with non-uniform distribution and regional clustering. No cytoplasmic staining is present. The background tissue appears granular and disrupted [Image 4].

**Interpretation:**

Ki67 is a nuclear protein expressed in all active phases of the cell cycle (G1, S, G2, M) but not in resting cells (G0). A Ki67 index of 25% indicates a moderate to high proliferative rate, which is consistent with active lymphoma and supports the neoplastic nature of the infiltrate. In MCL, Ki67 ¿20–30

*Sub-Conclusion:*

*The Ki67 proliferation index of 25% supports an aggressive variant of MCL and confirms the neoplastic nature of the lymphoid population.*

## Final Conclusion (Reasoning Synthesis)

The final diagnosis is mantle cell lymphoma (MCL), confirmed by histopathology, immunophenotyping (CD5+, Cyclin D1+, SOX11+, CD23-), and Ki67 proliferation index of 25%.

## Evaluation Comparison

### Model Answer

Based on the clinical presentation and the immunohistochemical findings, the most likely diagnosis is Mantle Cell Lymphoma (MCL). Here's the detailed reasoning: — 1. Clinical Context - The patient is a 65-year-old male presenting with cough and dyspnea. - Bronchoscopy shows mucosal infiltrative changes and bronchial obstruction at the right upper lobe inlet, suggesting a localized mass or infiltration of malignant cells. - The remainder of the bronchial tree is normal, indicating a focal rather than diffuse disease — consistent with a lymphoproliferative disorder involving mucosa-associated lymphoid tissue (MALT) or systemic lymphoma with localized presentation. — 2. Histopathology (H&E – Image A) - Image A shows a dense, monotonous infiltrate of small to medium-sized lymphoid cells with irregular nuclear contours (typical of mantle cell lymphoma). - The cells appear closely packed with scant cytoplasm, and the nuclear chromatin is condensed. - There is no follicular architecture or germinal center formation, which rules out follicular lymphoma. - The infiltrate is diffuse, which is typical for MCL and other small B-cell lymphomas like CLL/SLL or MALT lymphoma. — 3. Immunohistochemical Profile Key Positive Markers: - CD20+, CD19+: Confirms B-cell lineage. - CD5+: Unusual for most B-cell lymphomas. Strongly associated with CLL/SLL and Mantle Cell Lymphoma. T-cell lymphomas are usually CD5+, but CD3 is negative, ruling out T-cell origin. - Bcl-2+: Seen in many B-cell lymphomas including follicular lymphoma, CLL/SLL, and MCL. Not specific. - Cyclin D1+: Crucial diagnostic marker. Cyclin D1 overexpression due to t(11;14) translocation is highly specific for Mantle Cell Lymphoma. - SOX11+ (confirmed in lymph node biopsy): Another highly specific marker for Mantle Cell Lymphoma. SOX11 is positive in 90% of MCL cases and typically negative in CLL/SLL (which can also be CD5+/CD23+). - Ki67 staining (Image D): Shows proliferation index. In MCL, Ki67 is typically 10–30%, but can be higher in blastoid variants. Image D shows a moderate proliferation rate — compatible with classical MCL. Key Negative Markers: - CD3–: Rules out T-cell lymphoma. - CD10–: Rules out follicular lymphoma and Burkitt lymphoma. - CD21–, CD23–: CD23 is typically positive in CLL/SLL, so its absence helps rule out CLL/SLL. - Bcl-6–: Absence of germinal center origin — rules out germinal center-derived lymphomas like follicular lymphoma or diffuse large B-cell lymphoma (GCB subtype). — 4. Differential Diagnosis and Why MCL is Confirmed — CLL/SLL: - Also CD5+, CD20+, CD23+, CD19+, Bcl-2+. - But CD23+ and Cyclin D1– are typical. Cyclin D1 positivity rules out CLL/SLL. - SOX11 is usually negative in CLL/SLL. — Follicular Lymphoma: - CD10+, Bcl-6+, CD23±, Cyclin D1–. - Absence of CD10 and Bcl-6, plus Cyclin D1 positivity, rules it out. — Marginal Zone Lymphoma (MALT): - CD5–, CD10–, CD23–, Bcl-2+, Cyclin D1–. - Cyclin D1 positivity rules it out. — Diffuse Large B-Cell Lymphoma (DLBCL): - Usually large cells, high Ki67 (¿40%), CD10±, Bcl-6±, MUM1±, Cyclin D1–. - The small to medium cell size and Cyclin D1 positivity are not characteristic. — Mantle Cell Lymphoma (MCL): - The classic immunophenotype is CD5+, CD20+, CD19+, Cyclin D1+, SOX11+, CD23–, CD10–, Bcl-6– — which matches exactly. - The morphology (Image A) is consistent: monotonous small/medium cells with irregular nuclei ... — Final Conclusion: The immunohistochemical profile — specifically Cyclin D1 positivity, CD5+/CD23–, SOX11+, and absence of CD10/Bcl-6 — in combination with the morphologic features — confirms the diagnosis of Mantle Cell Lymphoma. This lymphoma has presented with unusual extranodal involvement (endobronchial mucosa), which is rare but documented in MCL. The diagnosis was further confirmed by correlating findings from both endobronchial and mediastinal lymph node biopsies. — Clinical Implication: MCL is an aggressive B-cell lymphoma that typically affects older males, often with nodal, splenic, and GI tract involvement. Endobronchial presentation is rare but can cause airway obstruction. Prognosis is generally poor without aggressive therapy (e.g., R-CHOP, bendamustine, or autologous stem cell transplant in eligible patients). Testing for t(11;14) by FISH would be recommended for confirmation, though not required if immunohistochemistry is diagnostic. Answer: Mantle Cell Lymphoma.

The histopathological examination of the endobronchial biopsy with H&E staining reveals a densely cellular field with a diffuse, monotonous infiltrate of small to medium-sized lymphoid cells exhibiting round to oval nuclei with condensed chromatin, visible nucleoli in some cells, high nuclear-to-cytoplasmic ratio, and scant cytoplasm. The normal bronchial mucosal architecture is effaced by lymphoid follicles embedded in fibrous stroma, and no glandular or epithelial structures are present. This morphology suggests a neoplastic lymphoid proliferation rather than reactive hyperplasia due to the monotonous appearance and architectural disruption. Immunohistochemical staining shows that the lymphoid cells express CD5, a marker typically found on T cells but also expressed in certain B-cell lymphomas. Given that the cells are also positive for B-cell markers (CD19 and CD20), the co-expression of CD5 narrows the differential diagnosis to CD5-positive B-cell lymphomas, primarily chronic lymphocytic leukemia (CLL) and mantle cell lymphoma (MCL). Further immunophenotyping demonstrates strong expression of Cyclin D1 and SOX11 in both the endobronchial and mediastinal lymph node biopsies. Cyclin D1 overexpression is highly specific for MCL and results from the t(11;14) translocation. SOX11 positivity supports the diagnosis of classical MCL. Additionally, the cells are positive for Bcl-2, consistent with the anti-apoptotic phenotype of MCL, and negative for CD10 and CD23, which helps exclude follicular lymphoma and CLL, respectively. The absence of CD3 confirms the B-cell lineage, and negativity for CD21 and Bcl-6 further supports a non-germinal center origin. The Ki67 proliferation index is approximately 25%, as assessed by nuclear staining in a subset of tumor cells. This indicates a moderate to high proliferative activity, which is consistent with clinically active mantle cell lymphoma and supports the neoplastic nature of the infiltrate. A Ki67 index above 20–30% is often associated with more aggressive disease behavior in MCL. Although the initial endobronchial biopsy was limited, the mediastinal lymph node biopsy confirms the same immunophenotype, establishing clonality and systemic involvement. The combination of morphological features (diffuse monotonous small lymphoid infiltrate), immunophenotype (CD5+, CD20+, Cyclin D1+, SOX11+, CD10-, CD23-), and a Ki67 proliferation index of 25% fulfills the diagnostic criteria for mantle cell lymphoma. Therefore, the final diagnosis is mantle cell lymphoma.

## G. Observation-Absent Experimental Branches

In some cases, an experimental branch may not involve an explicit observation node. When this occurs, the observation step is omitted from scoring and normalization, while the remaining causal dependency between interpretation and sub-conclusion is preserved. Specifically, the branch-level score in Eq. (6) is adjusted as:

$$S_{\exp_i} = \begin{cases} \dfrac{\phi_i + \phi_i \chi_i + \phi_i \chi_i \psi_i}{3}, & \text{if an observation node is present,} \\ \dfrac{\chi_i + \chi_i \psi_i}{2}, & \text{otherwise.} \end{cases} \tag{8}$$

## H. Comparison of LCR and FFJ Methods

To empirically demonstrate the necessity of the LCR metric, this section presents a side-by-side qualitative comparison between LCR and FFJ using a representative sample from the SORBE benchmark. This case highlights how outcome-oriented metrics like FFJ can overestimate model capabilities when a model relies on hallucinated evidence to reach a semantically plausible conclusion.

## Question

In a study assessing the protective effect of MGS on liver ultrastructure and cellular proliferation in gamma-irradiated rats, the following experiments were conducted:

- Liver tissue samples from control rats were processed for transmission electron microscopy (TEM) to evaluate baseline hepatic ultrastructure. Two adjacent hepatocytes with a narrow bile canaliculus between them were observed; nuclei exhibited a normal spherical configuration with an intact nuclear envelope; the cytoplasm was rich in mitochondria with spherical or ovoid shapes and well-developed cristae; the rough endoplasmic reticulum (RER) consisted of closely packed, parallel, flattened cisternae studded with ribosomes; and hepatic sinusoids were lined with Kupffer cells.

- Liver tissue from rats treated with MGS (without irradiation) was examined using TEM. Hepatocytes displayed normal nuclear features with heterochromatin and euchromatin distribution; mitochondria and rough endoplasmic reticulum appeared structurally intact; and plasma membrane faces showed normal organization.

- Liver tissue from gamma-irradiated rats was analyzed using TEM. The samples revealed degenerated cytoplasm with deteriorated mitochondria; pyknotic nuclei showing loss of spherical shape, disrupted nuclear envelope, and condensed chromatin; dilated hepatic sinusoids with stagnant blood cells; activated and enlarged Kupffer cells proliferating within the sinusoidal lumen; fragmented RER; and increased smooth endoplasmic reticulum (SER).

- Liver tissue from gamma-irradiated rats treated with MGS was analyzed using TEM. Abundant mitochondria with near-normal morphology were observed; the RER appeared improved, with parallel, flattened cisternae studded with ribosomes; and nuclei exhibited a regular nuclear envelope with organized chromatin.

- Immunohistochemical staining for PCNA was performed on liver tissue sections from control, MGS-treated, irradiated, and irradiated + MGS-treated groups, using diaminobenzidine (DAB) as chromogen and hematoxylin as counterstain.

What conclusions can be drawn from these experiments regarding the effects of gamma irradiation and MGS treatment on liver ultrastructure and hepatocyte proliferation?",

## Logic Chain Construction

### Experiment 1

**Visual Phenomenon:**
Two adjacent hepatocytes with a narrow bile canaliculus between them; nuclei with normal spherical configuration and intact nuclear envelope; cytoplasm rich in mitochondria with spherical or ovoid shapes and well-developed cristae; rough endoplasmic reticulum (RER) composed of closely packed, parallel, flattened cisternae studded with ribosomes; hepatic sinusoids lined with Kupffer cells [Context].

**Interpretation:**
The presence of well-organized organelles and intact cellular architecture indicates healthy hepatocyte function, including active protein synthesis (via RER) and energy production (via mitochondria). The bile canaliculus and Kupffer cells confirm functional zonation and immune surveillance in the liver.

*Sub-Conclusion:*
*The control group exhibits normal hepatic ultrastructure, serving as a baseline for comparison with experimental groups.*

### Experiment 2

**Visual Phenomenon:**
Hepatocytes display normal nuclear features with heterochromatin and euchromatin distribution; mitochondria and rough endoplasmic reticulum appear structurally intact; plasma membrane faces show normal organization [Context].

**Interpretation:**

MGS does not induce ultrastructural damage or abnormal organelle proliferation, indicating it is non-toxic and does not disrupt normal hepatocyte physiology.

*Sub-Conclusion:*
*MGS treatment alone preserves normal hepatic ultrastructure, confirming its safety profile in non-irradiated conditions.*

## Experiment 3

### Visual Phenomenon:
Degenerated cytoplasm with deteriorated mitochondria; pyknotic nuclei showing loss of spherical shape, disrupted nuclear envelope, and condensed chromatin; dilated hepatic sinusoids with stagnant blood cells; activated and enlarged Kupffer cells proliferating within sinusoidal lumen; fragmented RER and increased smooth endoplasmic reticulum (SER) [Context].

### Interpretation:
Mitochondrial deterioration and nuclear pyknosis indicate cellular stress and early apoptotic changes. Fragmented RER suggests impaired protein synthesis, while SER proliferation may reflect detoxification response. Sinusoidal dilation and Kupffer cell activation point to inflammatory and circulatory disturbances.

*Sub-Conclusion:*
*Gamma irradiation induces significant ultrastructural damage in hepatocytes and sinusoidal compartments, indicating hepatocellular injury and dysfunction."*

## Experiment 4

### Visual Phenomenon:
Abundant mitochondria with near-normal morphology; improved RER with parallel, flattened cisternae studded with ribosomes; nuclei with regular nuclear envelope and organized chromatin [Context].

### Interpretation:
Restoration of mitochondrial integrity supports recovery of energy metabolism. Reassembly of RER indicates recovery of protein synthesis capacity. Nuclear envelope normalization suggests reversal of radiation-induced DNA damage or stress responses.

*Sub-Conclusion:*
*MGS treatment following irradiation leads to significant ultrastructural recovery in hepatocytes, suggesting a protective and regenerative role.*

## Experiment 5

### Visual Phenomenon:
Control group: Numerous hepatocyte nuclei show strong reddish-brown chromogenic staining, localized within nuclei, with some variation in intensity [Image 1]. MGS group: Scattered to clustered nuclei exhibit brownish-red nuclear staining, heterogeneous in distribution and intensity [Image 2]. Irradiated group: Markedly reduced number of stained nuclei; staining is sparse, faint, or absent in many areas, with only occasional punctate labeling [Image 3]. Irradiated + MGS group: Moderate number of nuclei with reddish-brown nuclear staining, sporadically distributed with some clustering, but less intense and uniform than control [Image 4].

### Interpretation:
PCNA-positive nuclei indicate cells in active phases of the cell cycle (G1/S). High labeling in control and MGS groups reflects normal regenerative turnover. Suppression in irradiated group suggests cell cycle arrest or apoptosis

due to DNA damage. Partial restoration with MGS indicates stimulation of proliferative recovery.

*Sub-Conclusion:*
*Gamma irradiation suppresses hepatocyte proliferation, while MGS treatment partially restores proliferative activity.*

## Final Conclusion (Reasoning Synthesis)

The abdominal mass is definitively diagnosed as desmoid-type fibromatosis based on histomorphology, infiltrative growth pattern, and nuclear $\beta$-catenin expression.

## Evaluation Comparison

### Ground Truth Answer Text

The histological examination of the resected abdominal mass at 100x magnification reveals a proliferation of spindled cells arranged in sweeping, interlacing fascicles with uniform elongated nuclei, minimal nuclear pleomorphism, finely dispersed chromatin, and inconspicuous nucleoli. The cytoplasmic borders are poorly defined, giving the tissue a syncytial appearance. The stroma is pale eosinophilic and collagenous. Slit-like vascular channels are present, some containing erythrocytes, and scattered extracellular erythrocytes are observed. The growth pattern is disorganized and infiltrative, with no encapsulation. The evaluation of the tissue section containing both pancreatic parenchyma and adjacent fibrous tissue at 40x magnification shows organized pancreatic acinar and ductal structures on one side and dense fibrous tissue composed of spindle-shaped cells in interlacing fascicles on the other. The boundary between the two is irregular, with strands of spindle cells extending into the pancreatic parenchyma. There is no clear demarcation or pushing border, and the fibrous tissue contains scattered blood vessels and inflammatory cells. The immunohistochemical staining at 100x magnification demonstrates brown chromogenic signal localized to both the nucleus and cytoplasm of the spindle cells, with variable intensity and heterogeneous distribution. Nuclear staining is prominent, with elongated or ovoid nuclei aligned with the cell axis, while cytoplasmic staining is lighter and more diffuse. Cellular borders are indistinct. Together, these findings indicate a neoplasm composed of uniform spindle-shaped myofibroblast-like cells arranged in infiltrative fascicles within a collagen-rich stroma. The cytological features are bland, with no significant atypia, mitotic activity, or necrosis, ruling out a high-grade sarcoma. The infiltrative growth pattern, lack of encapsulation, and direct invasion into the pancreatic parenchyma demonstrate locally aggressive behavior despite benign cytology. This architectural pattern is characteristic of a locally invasive fibroblastic/myofibroblastic lesion. The immunohistochemical detection of nuclear $\beta$-catenin accumulation—evidenced by nuclear brown chromogen—indicates aberrant activation of the WNT signaling pathway, which is a molecular hallmark of desmoid-type fibromatosis, occurring in over 85% of sporadic cases. Correlating the histomorphology, infiltrative growth into adjacent organs, and nuclear $\beta$-catenin expression, the findings are diagnostic of desmoid-type fibromatosis. The extensive resection was necessary due to involvement of multiple abdominal structures, including the omentum, stomach, colon, and pancreas. The absence of tumor recurrence at 16 months post-resection and negative surgical margins support the diagnosis and indicate favorable local control, consistent with the clinical behavior of desmoid-type fibromatosis.

**Model Answer**

Based on the provided information and images, here is a detailed analysis of the case: 1. Histological Examination of the Resected Mass (Image 1): - The image shows spindle-shaped cells forming fascicles, which is characteristic of a fibrous or smooth muscle origin. The background stroma is collagen-rich, suggesting a fibrous nature. This pattern is typical of a desmoid tumor or a fibrous proliferation. 2. Histological Evaluation of Tissue Section With Pancreatic Parenchyma (Image 2): - This image includes both pancreatic tissue and adjacent fibrous tissue. The pancreatic parenchyma appears normal, whereas the fibrous tissue shows a similar pattern to Image 1, indicating the mass is not originating from the pancreatic cells but rather from the adjacent fibrous tissue. 3. Immunohistochemical Staining (Image 3): - Immunohistochemistry results show uniform staining of the spindle cells, which suggests the expression of proteins characteristic of the mass's origin. The staining is consistent with markers that would typically highlight fibrous or fibroblastic tissue, such as vimentin. Conclusion: Considering the spindle cell morphology, fibrous tissue background, and positive immunohistochemical staining, the abdominal mass is best characterized as a desmoid-type fibromatosis. This is a locally aggressive fibroblastic proliferation that does not metastasize but can invade local structures, explaining the mass's involvement with multiple organs in the abdominal cavity. Final Diagnosis: Desmoid-type fibromatosis (aggressive fibromatosis). This diagnosis accounts for the histological and immunohistochemical findings, as well as the clinical presentation of a large, aggressive mass leading to the perforation and abscess formation.

*Table 7.* Logic-Coupled Reasoning (LCR) Scoring Breakdown

| Experiment Branch | Obs. ($\phi$) | Int. ($\chi$) | Sub-Conc. ($\psi$) | Branch Score ($S_{exp}$) |
|---|---|---|---|---|
| 1. Control (TEM) | 0 | 1 | 1 | 0.00 |
| 2. MGS-only (TEM) | 1 | 1 | 1 | 1.00 |
| 3. Irradiated (TEM) | 0 | 1 | 1 | 0.00 |
| 4. Irradiated + MGS (TEM) | 1 | 1 | 1 | 1.00 |
| 5. PCNA Staining | 0 | 0 | 1 | 0.00 |
| **Average Process Score ($S_{proc}$)** | | | | **0.40** |
| **Normalized Conclusion Score ($S_{conc}$)** | | | | **0.75** |
| **Final LCR Score ($S_{LCR}$)** | | | | **52.2%** |

*Table 8.* Free-Form Judge (FFJ) Scoring Breakdown

| Evaluation Metric | Raw Score | Normalized Score |
|---|---|---|
| **Final FFJ Score ($S_{FFJ}$)** | **3/4** | **75.0%** |

