# OpenReview forum: "Benchmarking the Scientific Mind: A Pathology-Derived Biomedical VQA Benchmark for Complex Scientific Reasoning"
_ICML.cc/2026/Conference — ICML 2026 regular_

### Official Review · Reviewer_metr · 2026-02-27

**Soundness:** 3
**Presentation:** 4
**Significance:** 3
**Originality:** 3
**Overall Recommendation:** 4
**Confidence:** 4

**Summary:**

This paper introduces SORBE, a benchmark designed to assess the reasoning capabilities of MLLMs in the pathology domain. To enable evaluation of MLLMs' reasoning ability, the authors propose a Logic-Coupled Reasoning (LCR) score, which uses a process-oriented metric to evaluate the alignment between visual observations, scientific interpretations, and conclusions. The dataset is constructed using an autonomous framework that distills experimental reasoning paths via a multi-expert (LLMs) consensus and verification pipeline. The authors provide a comprehensive evaluation of both proprietary (e.g., GPT-4o, Gemini 2.5) and open-source models, revealing a significant performance gap in systematic experimental reasoning.

**Compliance With Llm Reviewing Policy:**

Affirmed.

**Final Justification:**

This paper introduces SORBE, a benchmark aimed at evaluating the reasoning capabilities of MLLMs in pathology, together with a novel evaluation pipeline for reasoning assessment. The paper is clearly written, the experiment is thorough, and the reported results are convincing.

My only concern is that the paper somewhat overclaims its broader impact on the biomedical domain, given that the benchmark and evaluation are centered on pathology data. The contribution is valuable, but the framing should be better aligned with the actual empirical scope of the work.

Overall, however, I believe the strengths outweigh this weakness, and I remain in my support for acceptance.

**Key Questions For Authors:**

* Why does the author choose pathology data (PathCap) specifically?

**Limitations:**

Yes

**Strengths And Weaknesses:**

**Strength**
* **Novel Evaluation Metric:** The proposed LCR score is a meaningful and fairly novel contribution, especially the Reasoning Process Score that aggregates binary correctness signals using an AND logic across steps. This design helps distinguish SORBE from standard medical VQA benchmarks by going beyond answer-only evaluation and formalizing the ground truth as a structured sequence.
* **Methodological clarity:** The paper is well written and clearly presented. The multi-stage dataset construction pipeline and the rationale behind the scoring design are articulated clearly, making the overall methodology easy to follow and straightforward to implement in principle.

**Weakness**
* **Limited Domain Diversity:** Although positioned as a **biomedical** VQA benchmark, SORBE relies solely on the PathCap dataset, a histopathology dataset. This narrow scope limits the benchmark’s generality in assessing reasoning ability within other biomedical imaging modalities (e.g., CT, MRI, chest X-ray, etc.) and the breadth of conclusions that can be drawn.
* **Potential Evaluator Bias from LLM-as-the-judge:** Since this work is positioned as a benchmark where the "judge" serves as a ground-truth oracle, a more rigorous validation of the automated evaluation framework is necessary. Specifically:
    * **Validation Against Oracles/Reference set:** The reliance on an LLM-based verifier to score complex, multi-step scientific logic chains introduces a risk of bias or self-preference [1,2]. The authors do not provide a correlation analysis between the LLMs and expert human judgment, only the quality control for dataset curation. Without this sanity check, it is difficult to determine if the reported performance degradation on SORBE reflects genuine model failure or is simply an artifact of the evaluator's inability to parse specialized biomedical nuances. As shown in Table 6, the SORBE score differs significantly across judge models, e.q, Qwen vs DeepSeek.
   * **Sensitivity to Prompting:** The paper provides a comprehensive prompt library, but it does not discuss how sensitive the LCR score is to the specific prompts used by the evaluator. If a different prompt were used, it is unclear if the performance gap between models would remain consistent.

**Minor**
* Reproducibility: The dataset curation process employs an ensemble of four distinct MLLMs (experts) and a refiner (Qwen3-235B). This multi-expert consensus approach is computationally intensive, requiring significant GPU resources, which may pose a barrier for researchers attempting to replicate the construction pipeline or scale it to new domains.
* Missing details: In the method section, the author only describes the judge as a "constrained verifier" but does not specify the model architecture that the judge model used.



[1] Szymanski, Annalisa, et al. "Limitations of the llm-as-a-judge approach for evaluating llm outputs in expert knowledge tasks." Proceedings of the 30th international conference on intelligent user interfaces. 2025.

[2] Panickssery, Arjun, Samuel Bowman, and Shi Feng. "Llm evaluators recognize and favor their own generations." Advances in Neural Information Processing Systems 37 (2024): 68772-68802.

---

> ### Author Rebuttal · Authors · 2026-03-30
>
> We are truly grateful for the reviewer's valuable suggestions that helped refine our work.
>
> **Limited Domain Diversity (W1):** We candidly acknowledge in Section 5 that the Path-Cap data exhibits a phenomenon of concentration; however, SORBE is not limited to the field of histopathology. As evidence, we provide **Figure 3**(https://anonymous.4open.science/r/rebuttal_image-7527/) demonstrating complex reasoning chains built upon macro-radiological imaging, explicitly including CT, MRI, and X-rays.
>
> **Case 1: MRI-based Multi-modal Reasoning (Spinal Cord Pathology)**
>
> - Input Modalities: T2-weighted midsagittal MRI, post-contrast T1-weighted MRI, CSF laboratory analysis, and H&E histopathology.
>
> - Reasoning Requirement: The task requires the model to integrate **MRI signal intensities and enhancement patterns** with CSF protein levels (72.01 mg/dL) to differentiate between neoplastic and inflammatory etiologies.
>
> **Case 2: CT-based Multi-source Integration (Nasolacrimal Duct Pathology)**
>
> - Input Modalities: Dacryocystography (frontal radiography), **axial CT scan**, nasal endoscopy, and surgical histopathology.
>
> - Reasoning Requirement: The model must analyze **space-occupying lesions in the nasal cavity via CT scans** and coordinate these with dacryocystography findings and pancytokeratin expression to reach a definitive diagnosis.
>
> These examples demonstrate that SORBE’s reasoning depth extends across the full clinical diagnostic workflow, explicitly incorporating **CT, MRI, and X-ray** modalities beyond histopathology.
>
> Methodological Scalability: Our agentic workflow for logic chain construction is inherently domain-agnostic. The underlying framework for extracting causal reasoning from scientific literature can be seamlessly deployed to any biomedical discipline beyond pathology.
>
> **Potential Evaluator Bias from LLM-as-the-judge(W2):** To address concerns regarding evaluator bias, we emphasize that LCR effectively eliminates subjectivity via strict atomic semantic alignment, an objective mechanism we empirically validate below through cross-evaluator consistency (W2.1) and rank-preserving prompt robustness (W2.2).
>
> **Validation Against Oracles/Reference set (W2.1)**: To rigorously validate our automated evaluation framework and rule out self-preference, we conducted a comprehensive cross-evaluator consistency experiment. We selected five diverse, state-of-the-art LLMs—varying significantly in scale (14B to 235B) and architecture—to serve as LCR judges for an identical set of 3,799 Gemini 2.5 Flash responses.
>
> **Tab C: Cross-Evaluator Consistency (LCR)**
>
> | **Evaluator Model**    | **LCR**   |
> | ---------------------- | --------- |
> | **Qwen3-235B**         | **42.12** |
> | DeepSeek-V3.2          | 41.59     |
> | MiniMax-2.5            | 41.00     |
> | Qwen3-14B              | 40.79     |
> | Qwen3-30B-A3B-Instruct | 40.76     |
>
> The scoring consistency across five diverse judges (variance of 0.35) demonstrates that LCR's atomic semantic verification effectively eliminates subjective evaluator bias. Consequently, the performance degradation in SORBE is an objective reflection of genuine multi-step reasoning deficits in current models.
>
> **Sensitivity to Prompting (W2.2)**
>
> We acknowledge that absolute LCR scores are naturally sensitive to the stringency of prompt instructions. To evaluate the impact on the metric's reliability, we conducted an ablation study on a fixed subset (n=100) using three prompt variations with varying levels of strictness.
>
> **Tab D: Impact of Prompt Strictness on LCR Scores**
>
> | **Evaluation Prompt Strictness** | **LCR** | **Std_Dev** |
> | -------------------------------- | ------- | ----------- |
> | Prompt A (Lenient)               | 57.34   | 33.76 |
> | Prompt B (Standard)              | 53.97   | 33.23 |
> | Prompt C (Strict)                | 45.12   | 33.13  |
>
> **Tab E: Spearman Rank Correlation Matrix across Prompt Settings**
>
> ||Lenient|Standard|Strict|
> |---|---|---|---|
> |Lenient|1.000|0.744|0.712|
> |Standard|0.744|1.000|0.694|
> |Strict|0.712|0.694|1.000|
>
> While prompt stringency predictably shifts absolute mean scores (global trend $\rho = -0.151$, $p < 0.01$), intra-batch variance remains stable ($\approx 33$). Strong intra-model Spearman rank correlations ($\rho \ge 0.694$, Tab E) confirm the consistent penalization of flawed reasoning, thereby preserving ordinal model rankings and relative performance gaps. This demonstrates that prompt variations merely adjust the evaluation baseline without compromising LCR's discriminative power, validating it as a robust metric for comparative assessment under consistent prompting. I will discuss this issue in detail in the revision.
>
> **Reproducibility (M1)**: The multi-expert ensemble incurs a one-time computational cost. Subsequent LCR evaluations require only standard, efficient inference.
>
> **Judge Architecture (M3)**: Section 4.4 now specifies the primary judge is DeepSeek-V3.2.

---

> > ### Author Rebuttal · Reviewer_metr · 2026-03-31
> >
> > Thank you for your responses. I will maintain my score and remain on the acceptance side. However, I do think the author overclaims its contribution to the biomedical domain, given the limited domain diversity in the evaluated dataset. This limitation should be stated explicitly upfront, particularly in the title, abstract, and introduction.

---

> > > ### Author Response · Authors · 2026-04-01
> > >
> > > We sincerely thank you for your continued support, constructive feedback.
> > >
> > > We agree with your assessment regarding the scope of our claims. To ensure scientific rigor and avoid overclaiming, we will explicitly state this domain limitation upfront in the camera-ready version. Specifically, we will revise the contribution statement in the introduction to accurately reflect our current. Furthermore, we will discuss our plans to validate the SORBE framework across broader biomedical modalities in our future work.

---

### Official Review · Reviewer_8b3y · 2026-03-09

**Soundness:** 3
**Presentation:** 4
**Significance:** 3
**Originality:** 3
**Overall Recommendation:** 3
**Confidence:** 3

**Summary:**

The paper introduces SORBE (Scientific Observation & Reasoning for Biomedical Evaluation), a novel multi-image biomedical Visual Question Answering (VQA) benchmark. The authors argue that existing benchmarks fail to adequately capture multi-step scientific reasoning because they rely on single images and outcome-oriented evaluation. To address this, the authors propose a data construction framework that uses a Multi-Expert Visual Integration module to extract visual evidence , generates structured logic chains , and applies a Logic-Coupled Reasoning (LCR) score that evaluates both the final conclusion and the intermediate reasoning process. Empirical results show that current state-of-the-art Multimodal Large Language Models (MLLMs) struggle significantly on this benchmark compared to standard medical VQA tasks.

**Compliance With Llm Reviewing Policy:**

Affirmed.

**Key Questions For Authors:**

See weaknesses

**Limitations:**

Yes

**Strengths And Weaknesses:**

Strengths:

1.	Strong Motivation: The paper addresses a critical flaw in existing benchmarks by shifting the evaluation paradigm from outcome-oriented (pattern matching) to process-oriented (evidence-driven reasoning).
2.	Robust Evaluation Metric: The proposed Logic-Coupled Reasoning (LCR) score effectively penalizes process hallucinations. By enforcing cascading penalties, it prevents models from gaming the metric by arriving at a correct conclusion through flawed intermediate reasoning.
3.	Rigorous Data Curation Pipeline: The automated multi-expert consensus protocol for visual integration and the multi-round quality control (QC) for logic chains ensure a high-quality, hallucination-free ground truth.

Weaknesses:
1. Data Source Homogeneity: The paper claims to evaluate broad biomedical reasoning , yet the benchmark is exclusively derived from the PathCap dataset (pathology/histology images). This single-domain source overstates the benchmark's generalizability and fails to cover other critical biomedical modalities (e.g., radiology, genomics charts).
2. Homogenization Bias in LLM-as-a-Judge: The evaluation heavily relies on an LLM judge (DeepSeek-V3.2) to evaluate other LLMs. It completely lacks a human-LLM agreement analysis (e.g., alignment with domain experts), making it difficult to determine whether low scores stem from true model failures or the inherent biases/strictness of the evaluator LLM.
3. Absence of Frontier Reasoning Models: Despite targeting "complex scientific reasoning," the experiments rely on general-purpose or speed-optimized models (e.g., GPT-4o, Claude 4 Sonnet, Gemini-2.5-Flash). The exclusion of flagship reasoning-focused models (such as the OpenAI o-series, Claude 4 Opus, or Gemini 2.5 Pro) fails to probe the true ceiling of current AI capabilities, thereby weakening the core claim regarding the systematic limitations of existing models.

---

> ### Author Rebuttal · Authors · 2026-03-30
>
> We sincerely thank the reviewers for their professional insights and suggestions, which will be highly instrumental in helping us enhance the quality of our manuscript.
>
> **Data Source Homogeneity (W1)**: As we transparently acknowledged in Section 5 (Limitations), the current dataset has a concentration in certain domains like pathology.
>
> However, we took active measures to mitigate this. During the data generation phase, we introduced four domain-specific keywords—Basic Medicine, Clinical Medicine, Diagnostic and Laboratory Medicine, and Pharmacy and Therapeutics (see Appendix A.2)—aiming to infuse the question-answer pairs with a broader biomedical context.
>
> More importantly, the core contribution of this study—namely, the "data-generating agent workflow"—is highly scalable and domain-agnostic. We directly applied our pipeline to the med-frameqa dataset (which focuses on Radiology/Imaging). Our framework successfully and automatically generated high-quality logic chains for these new modalities.
>
> Due to space constraints, we provide a condensed snippet of a newly generated logic chain for a lumbar spine MRI case:
>
> - Visual Phenomenon: A mass is observed compressing the right nerve root and displacing it laterally (Image 1 & 2).
>
> - Interpretation: Preserved perineural fat indicates the nerve root is pushed laterally but not completely encased, suggesting a foraminal lesion.
>
> - Sub-conclusion: The nerve root is laterally displaced by a contained foraminal disc herniation.
>
> **Homogenization Bias in LLM-as-a-Judge (W2):**
> We appreciate the reviewer's attention to the potential bias and strictness of the evaluator LLM. We wish to clarify our evaluation strategy and provide empirical evidence demonstrating that the low scores reflect actual model reasoning failures rather than evaluator strictness.
>
> **Mitigating Subjectivity via Logic Chains:** To minimize bias, LCR replaces open-ended LLM judgments with a strict comparison against predefined atomic facts in reference logic chains. This mechanism significantly suppresses the inherent subjective bias of the evaluator LLM.
>
> **Discerning Deficiencies from Strictness:** Analysis of 3,799 Gemini 2.5 Flash responses  confirms that errors primarily stem from logical flaws rather than evaluation harshness. The average Conclusion Score (61.6) is significantly higher than the Reasoning Score (41.8), proving the evaluator awards high scores when conclusions are correct.
>
> **Evidence of Shortcut Reasoning:** Among low-scoring samples (Total < 60), 40.7% (977 samples) successfully guessed the medical diagnosis (Conclusion $\ge$ 75) but failed in intermediate reasoning (Reasoning $<$ 50). This performance gap, visualized in **Figure 2**(https://anonymous.4open.science/r/rebuttal_image-7527/) by the massive cluster in the bottom-right corner, confirms that LCR scores are not artifacts of bias but accurately penalize "shortcut" guessing.
>
> **Concrete Case Study**: Consider a real pathology sample in our dataset involving a high-grade bladder/prostate tumor. The model generated an ostensibly professional and fluent lengthy response, detailing the tumor's aggressiveness and heterogeneity. Under traditional outcome-only evaluations, such responses easily secure high scores. However, the LCR mechanism assigned a critically low score. Cross-referencing the logic chain revealed that the model merely paraphrased the provided clinical context (e.g., vascular differentiation, high aggressiveness). It failed to extract decisive visual features from the medical image and did not derive the substantive final diagnosis (high-grade urothelial carcinoma with angiosarcomatoid differentiation), even resorting to speculative phrasing like "might show."
>
> **Absence of Frontier Reasoning Models (W3)**: We sincerely thank the reviewers for their suggestions. We agree that evaluating **flagship reasoning models** is essential to delineate the performance ceiling of MLLMs on complex scientific reasoning. While our initial experiments focused on the most accessible state-of-the-art models at the time of submission (e.g., GPT-4o, Gemini-2.5-Flash), we have since conducted **supplementary experiments** using **Gemini 2.5 Pro** to address this concern.
>
> We evaluated Gemini 2.5 Pro on a representative subset of **samples**. The model achieved an **LCR score of 48.5**. Although this achieved a performance improvement compared to "Flash," it still did not reach a high score. These results further validate **SORBE as a rigorous and challenging benchmark**, even for the most advanced reasoning-focused models. In the revision, we will include experiments using flagship models (Claude 4 Opus or OpenAI o1).

---

> > ### Author Rebuttal · Reviewer_8b3y · 2026-04-01
> >
> > The rebuttal partially addresses the reviewers’ concerns, but does not fully resolve them. For data source homogeneity, it shows that the generation workflow is extensible beyond pathology and provides a radiology example, yet this does not fully support the paper’s broader claim of benchmark generalizability because the current benchmark remains largely single-domain. For the LLM-as-a-judge concern, the response is more convincing: the authors clarify that scoring is anchored to predefined atomic facts rather than unconstrained judgment, and their error analysis suggests that low scores reflect genuine reasoning failures rather than simple evaluator harshness; however, the lack of human-expert agreement analysis remains an important gap. For the absence of frontier reasoning models, the additional Gemini 2.5 Pro result is helpful, but evaluation on only a representative subset and the promise of future experiments with stronger models still leave the performance ceiling insufficiently established.

---

> > > ### Author Response · Authors · 2026-04-01
> > >
> > > We thank you for your continued engagement and for acknowledging the strengths of our LCR metric. We appreciate the opportunity to address your remaining concerns with the comprehensive evidence below.
> > >
> > > **1. Data Source Homogeneity & Generalizability Claims:** To present the contributions of our paper more clearly and ensure scientific rigor, we will explicitly refine the scope of our claims in the camera-ready version. Specifically, we will revise the title or the introduction to accurately reflect that the current benchmark focuses on the pathology domain, rather than the broader biomedical field. Furthermore, our future work will explore strategies to validate the applicability of the SORBE framework across a wider range of biomedical modalities.
> > >
> > > **2. Human-Expert Agreement Analysis (Evaluator Bias):** Due to strict space constraints during the initial rebuttal phase, this analysis was previously omitted. As shown in our detailed response to Reviewer fsBs, we conducted a rigorous expert validation study to directly address the gap regarding human-LLM agreement. Biology PhD annotators blindly reviewed 200 divergent samples (high outcome-oriented scores but low LCR scores). The evaluation revealed that in 90.1% of these cases, the low LCR scores accurately reflect genuine factual reasoning errors (e.g., incorrect observations or invalid logic) rather than evaluator over-penalization. This empirical alignment confirms that LCR reliably measures true reasoning quality rather than merely applying harsher grading.
> > >
> > > **3. Establishing the Performance Ceiling with Frontier Models:** Due to API rate limits and access restrictions during the initial rebuttal phase, our evaluation of flagship models was temporarily constrained to a subset. Having resolved these access limitations, we have completed the full-set evaluation for three frontier reasoning models. The LCR scores on the complete benchmark are detailed in Table F:
> > >
> > > **Table F: Full-Set LCR Evaluation of Frontier Reasoning Models**
> > >
> > > | Model           | LCR Score | Remarks             |
> > > | :-------------- | :-------: | :------------------ |
> > > | Gemini 2.5 Pro  |   47.70   | Newly Added         |
> > > | Claude 4 Opus   |   32.83   | Newly Added         |
> > > | GPT 5 Mini      |   26.75   | Newly Added         |
> > > | Claude 4 Sonnet |   31.40   | Original Submission |
> > >
> > > The evaluation of these three newly added advanced models demonstrates that they still struggle to achieve high scores. This indicates that even for more advanced models, evidence-grounded reasoning remains flawed.
> > >
> > > We value your constructive feedback, which has significantly strengthened our paper. We hope these comprehensive evaluations resolve your remaining concerns, and merit a favorable reassessment of our submission.

---

### Official Review · Reviewer_QbGe · 2026-03-12

**Soundness:** 4
**Presentation:** 4
**Significance:** 3
**Originality:** 2
**Overall Recommendation:** 4
**Confidence:** 4

**Summary:**

The paper proposes a framework to construct biomedical VQA items that explicitly encode multi-step reasoning grounded in experimental evidence. It introduces SORBE, a multi-image benchmark produced from literature-derived image–caption sets using an LLM-driven pipeline that aggregates “expert” model observations and generates logic chains. A process-oriented metric, LCR, scores both the final conclusion and the reasoning steps via a harmonic mean that couples a conclusion score with a reasoning-process score based on binary, dependency-linked nodes per evidence branch. Experiments across proprietary and open-source MLLMs show large performance drops on SORBE compared with existing datasets; an ablation suggests LCR is stricter than a free-form judge.

**Compliance With Llm Reviewing Policy:**

Affirmed.

**Final Justification:**

This paper presents a meaningful step toward process-oriented evaluation in biomedical VQA by introducing the SORBE benchmark and the LCR metric, which explicitly couples answer correctness with reasoning fidelity. The focus on multi-image, evidence-grounded reasoning is well motivated and supported by informative empirical results that reveal clear gaps in current MLLMs. Importantly, the authors have addressed the main concerns in the rebuttal with substantial improvements, including a more objective and robust evaluation protocol that mitigates model-induced bias, expanded expert verification, and clearer positioning with respect to related work. These additions significantly strengthen the credibility and rigor of the benchmark. While some limitations remain (e.g., verification coverage and potential dataset construction biases), the work is technically solid and likely to serve as a valuable resource for future research in multimodal medical reasoning.

**Key Questions For Authors:**

Please refer to the Weaknesses section.

**Limitations:**

yes

**Strengths And Weaknesses:**

**Strengths:**
1. The LCR score connects conclusion accuracy with reasoning fidelity via the harmonic mean and the dependency-penalty branch scores, thereby preventing cases of “correct answers for the wrong reasons.”

2. Multi-image, evidence-centric tasks: SORBE emphasizes cross-image causal reasoning rather than attribute recall from a single image. Figure 1 contrasts outcome-oriented evaluation with SORBE’s process-oriented design and visually illustrates the goal of aligning the logical chain.

3. Informative empirical signals: Table 2 shows that models performing well on standard medical VQA benchmarks suffer dramatic drops on SORBE, indicating that SORBE tests a distinct set of capabilities. The breakdown in Table 3 further shows that conclusion scores are much higher than process scores, highlighting deficiencies in reasoning consistency.

**Weakness:**

1. The model uses Qwen3-VL, Lingshu-32B, HuluMed-32B, and Fleming-VL-38B to extract observational data, then performs reasoning with Qwen3-235B. Several model families are evaluated in Table 2. This dual role may introduce bias: if a model’s stylistic features align with or contradict the actual reasoning chains and the annotators’ expectations, it could gain an advantage or be disadvantaged. The paper does not analyze this confounding factor, nor does it report sensitivity across different annotators. This issue should be discussed.

2. Verifying only 10% of the items is insufficient, as it leaves the majority of the reasoning steps unchecked. Ideally, all items should be verified; sampling is acceptable, but it is difficult to determine an exact proportion. Intuitively, verifying more than 20% would likely be more convincing.

**Potential missing reference:**

1） "M3CoTBench: Benchmark Chain-of-Thought of MLLMs in Medical Image Understanding." ( ICLR 2026). Both the two work aim to overcome the limitations of traditional medical visual question answering, which focuses solely on outcomes. Their core goal is to evaluate the complex CoT and multi-step reasoning abilities of large multimodal models for medical image understanding. Of course, the focus is different. The advantage of SORBE lies in its large-scale data engineering: it constructs a larger dataset of 3,799 high-quality multi-image question–answer pairs and supports multi-image inputs. This work should also be included in Table 1.

2）"MME-CoT: Benchmarking Chain-of-Thought in Large Multimodal Models for Reasoning Quality, Robustness, and Efficiency." (ICML 2025) Although MME-CoT focuses on natural images, it is one of the earliest works to systematically transition from outcome-oriented evaluation to assessing the quality, robustness, and efficiency of the intermediate reasoning steps themselves.

---

> ### Author Rebuttal · Authors · 2026-03-30
>
> We are deeply grateful to the reviewers for raising these highly constructive and pertinent points. We will now attempt to address the issues raised by the reviewers.
>
> **Concerns regarding Model Bias and Evaluation Sensitivity (W1)**:First, we clarify that the vision expert models (e.g., Qwen3-VL, Lingshu-32B) are strictly limited to visual feature extraction tasks (e.g., outputting "spindle cells detected"). They are not involved in constructing logic chains or performing final diagnostic reasoning. Therefore, they do not introduce bias into the reasoning process itself.
>
> **Upgrading to an Objective Evaluation Mechanism**:We fully agree that using different LLMs as evaluators can introduce subjective scoring fluctuations due to their inherent stylistic preferences. To mitigate this potential bias, we developed a new scoring mechanism during the rebuttal period. This mechanism eliminates the subjective judgment of LLMs and adopts an atomic semantic alignment approach based on established medical knowledge graphs. The strategy involves three key steps:
>
> - **(1) Atomic Fact Decomposition:** We utilize LLMs strictly to decompose the model's response into indivisible, self-consistent atomic facts. During this process, ambiguous pronouns are explicitly resolved, and inherent causal logical relationships are strictly preserved.
>
> - **(2) Dual Objective Alignment:** Instead of relying on LLM evaluations, we assess each atomic fact using two objective dimensions:
>
>     - _Medical Concept Extraction (CUI Match Rate):_ We utilize the scispacy toolkit and the UMLS linker to extract standardized medical concepts, calculating a deterministic CUI match rate. This is rigorously defined as the proportion of CUIs present in the reference logic chain fact that are successfully matched within the CUI pool of the model's overall response.
>
>     - _Semantic Similarity:_ We employ the Qwen3-Embedding model to map the model-generated atomic fact (the best match) and the reference logic chain fact into high-dimensional dense vectors, calculating their mathematical cosine similarity.
>
> - **(3) Deterministic Scoring:** The final evaluation for each fact is computed using a deterministic mathematical formula: $\text{Fact Score} = (\text{CUI Match Rate} + \text{Semantic Similarity}) / 2$. By combining the CUI match rate and cosine similarity, this objective formula thoroughly circumvents potential biases introduced by the stylistic preferences of any single evaluator model.
>
> By combining the CUI match rate and cosine similarity, this objective formula thoroughly circumvents potential biases introduced by the stylistic preferences of any single evaluator model.
>
> **Robustness and Sensitivity Verification**: To validate the robustness of this new mechanism, we selected five different frontier models and re-evaluated 3,799 samples from Gemini 2.5 Flash.
>
> **Tab B: Cross-Evaluator Consistency**
>
> |     | Evaluator Model        |  LCR  |
> | :-: | :--------------------- | :---: |
> |  ➊  | Qwen3-235B             | 42.12 |
> |  ➋  | Qwen3-30B-A3B-Instruct | 40.76 |
> |  ➌  | Qwen3-14B              | 40.79 |
> |  ➍  | MiniMax-2.5            | 41.00 |
> |  ➎  | DeepSeek-V3.2          | 41.59 |
>
> As shown in Tab B, the variance in the revised LCR scores across evaluation models with varying architectures (e.g., dense vs. MoE) and from different developers is negligible (variance $< 1.5\%$). This empirical evidence demonstrates that the LCR metric captures rigorous, objective medical facts and semantic alignments rather than subjective model preferences. This reconstructed mechanism ensures that our benchmark measures the true biomedical reasoning fidelity of multimodal large models strictly and without bias.
>
> **Insufficient Verification (W2)**:We acknowledge that increasing the verification ratio enhances the persuasiveness of the benchmark. Conducting broader validation indeed further strengthens the credibility of our reasoning chain evaluation.
>
> **Expanded Expert Verification:** During the rebuttal phase, we expanded the scope of human expert verification to 15% of the total dataset (570 samples).
>
>  **Verification Stability:** Verification by biology experts confirms that, the proportion of high-quality data is 91.4%. The evidence provided by this expanded validation further confirms the robustness of our logic-chain-guided pipeline and demonstrates that LCR scores accurately reflect the true reasoning quality across the entire dataset.
>
> **Commitment:** We commit to conducting comprehensive expert manual verification of the entire benchmark dataset.
>
> **Missing References:** We added _M3CoTBench_ to the literature review and Table 1. We now explicitly contrast _M3CoTBench_ to highlight SORBE’s large-scale multi-image biomedical contributions, and cite _MME-CoT_ as a pioneering framework transitioning toward process-oriented evaluation of intermediate reasoning.

---

> > ### Author Rebuttal · Reviewer_QbGe · 2026-04-03
> >
> > Thank you to the authors for the detailed response. My concerns have been adequately addressed, and I will maintain my score.

---

> > > ### Author Response · Authors · 2026-04-05
> > >
> > > Thank you for reviewing our rebuttal and confirming the resolution of your concerns. The constructive feedback provided has been instrumental in refining our work. All modifications discussed during the rebuttal phase will be strictly integrated into the final manuscript. We thank you again for the time and effort dedicated to reviewing this paper.

---

### Official Review · Reviewer_fsBs · 2026-03-12

**Soundness:** 2
**Presentation:** 3
**Significance:** 3
**Originality:** 3
**Overall Recommendation:** 4
**Confidence:** 4

**Summary:**

This paper introduces SORBE, a benchmark for complex biomedical reasoning in VQA, together with a process-aware metric, LCR. The goal is to move beyond answer correctness toward multi-image, evidence-grounded, multi-step scientific reasoning through logic-chain-guided data construction and joint evaluation of reasoning process and final conclusion.

**Compliance With Llm Reviewing Policy:**

Affirmed.

**Final Justification:**

My main concerns were addressed in the rebuttal, and I raised my score to 4.

**Key Questions For Authors:**

1. How robust is LCR to alternative but scientifically valid reasoning paths that do not follow the benchmark’s reference template? This would clarify whether the benchmark measures reasoning quality or conformity to a preferred format.

2. To what extent does SORBE performance reflect multi-image scientific reasoning, rather than increased input length and cross-image alignment difficulty? For example, comparison against expert human ratings would help clarify whether lower scores under LCR reflect better faithfulness to reasoning quality, rather than simply harsher scoring.

3. Can the response provide stronger evidence that LCR is not only stricter than free-form judge, but also more accurate? Validation against expert human ratings would be especially helpful.

4. Can the response better isolate the effect of logic-chain-guided construction on benchmark quality?

**Limitations:**

Yes. The paper discusses limitations of the source data and benchmark scope.

One possible improvement is to more clearly distinguish claims supported by direct empirical evidence from those motivated mainly by design intuition, especially regarding the benchmark construction pipeline and the interpretation of LCR.

**Strengths And Weaknesses:**

Strengths:
1. The paper addresses an important problem. It highlights that strong answer accuracy in existing biomedical VQA benchmarks may not reflect faithful evidence-based reasoning.

2. The proposed benchmark design is well aligned with this motivation. The paper contributes a new data construction pipeline and a process-aware evaluation metric.

3. The empirical results support the main claim. The consistent gap between conclusion scores and process scores suggests that current models often produce plausible answers without equally grounded reasoning.



Weaknesses:
1. The benchmark may introduce reference-structure bias. Since evaluation is tied to a fixed Observation → Interpretation → Sub-Conclusion template, it is unclear how well SORBE handles alternative but valid reasoning paths. The evaluation may also be sensitive to answer style or verbosity.

2. Some key design choices are insufficiently validated. In particular, the contribution of logic-chain-guided construction is not clearly disentangled from other factors that may affect benchmark difficulty.

3. LCR appears stricter than free-form judging, but not necessarily more accurate. Lower scores under LCR suggest more conservative evaluation, but do not by themselves show better alignment with true reasoning quality.

4. The source data appears limited in scope, which may narrow the range of biomedical reasoning settings covered by the benchmark.

---

> ### Author Rebuttal · Authors · 2026-03-30
>
> We thank you for the constructive feedback and address your concerns in the following point-by-point responses.
>
> **Reference-Structure Bias (W1+Q1)**: Our LCR framework performs semantic fact verification rather than structural pattern matching. By extracting atomic knowledge from free-form outputs and mapping it directly to the reference logic chain, LCR effectively eliminates structural bias.
>
> To demonstrate this, we analyzed a high-scoring sample involving liver ultrastructure (TEM) and PCNA immunohistochemistry, where the reference logic chain is structured sequentially. However, Gemini 2.5 Flash generated a highly free-form, narrative-style response organized by biological themes (e.g., 1. Effects of gamma irradiation, 2. Protective role of MGS). It logically intertwined visual observations (e.g., _very few PCNA-positive (brown) nuclei_) with biological interpretations (e.g., _impaired regenerative capacity of the liver_) within a bulleted list, thus deviating significantly from our predefined structural template. Despite this deviation, LCR extracted and aligned the core facts, assigning a full score.
>
> **Does SORBE and LCR reflect scientific reasoning (Q2+Q3+W3)**: We agree that SORBE performance reflects multiple factors, but our goal is to expose limitations in evidence-constrained reasoning that plausibility-based metrics overlook. The performance gap is unlikely due to input length alone, as SORBE induces a clear reordering of model rankings (Tables 2-3) compared to baseline benchmarks. Furthermore, models maintain robust scores under unconstrained metrics (FFJ, Med-CMR) with the same inputs, indicating that increased length/alignment is not the primary bottleneck.
>
> Accuracy vs. Harshness (Validation):
>
> 1. **Automated Macro-Analysis:** Among 3,799 Gemini 2.5 Flash responses, 550 exhibited significant divergence ($\ge 80$ under FFJ but $< 50$ under LCR). Notably, in 88.2% (LCR conclusion score $\ge 3$) of these cases, models correctly guessed the final diagnosis despite flawed intermediate reasoning (Fig. 1(https://anonymous.4open.science/r/rebuttal_image-7527/)).
>
> 2. **Expert Human Validation:** We invited Biology PhD annotators sampled 200 divergent cases, finding that 90.1% of low LCR scores reflect factual reasoning errors (e.g., e.g., incorrect observations or invalid interpretations logic) rather than over-penalization.
>
> An IMT case analysis further validates this alignment between LCR and human judgment:
>
> - **FFJ's Outcome Bias:** Erroneously assigned a high score because the model identified the correct diagnosis (IMT) and generated fluent justifications, failing to detect that the reasoning was unsupported by actual visual evidence.
>
> - **LCR's Atomic Verification:** Assigned 0/4 for visual phenomena (despite 4/4 for the conclusion), exposing hallucinations. The model missed the diagnostic fibrous capsule and ignored actual Actin/EMA positivity, instead generating markers based on incorrect prior knowledge.
>
> These results confirm that LCR better reflects faithfulness to reasoning quality, effectively correcting the "false positives" that outcome-biased judges (FFJ) systematically overlook. We will include this human-alignment analysis in the revision.
>
> **The contribution of logic-chain-guided construction(W2+Q4)**:
>
> 1.  Logic-chain-guided construction is not simply make the task harder. By constraining question generation with explicit intermediate reasoning, it anchors each item to verifiable evidence and biologically plausible causal structure rather than answer plausibility alone. Our current evidence (Tab A) supports this view. Models still score relatively well under unconstrained metrics (FFJ, Med-CMR), suggesting that long-context understanding is not the only bottleneck. In contrast, scores drop sharply under LCR, which verifies atomic reasoning steps. This indicates that many responses are plausible in form but flawed in intermediate logic.
>
> **Tab A: Comparison of Different Evaluation Metrics**
>
> | Method  | Evaluation Paradigm             | Score |
> | :------ | :------------------------------ | :---: |
> | LCR     | Atomic Logic-Chain Verification | 40.34 |
> | FFJ     | Unconstrained Free-Form         | 68.00 |
> | Med-CMR | Medical QA Assessment           | 65.64 |
>
> 2.  In the revision, we will clarify that logic-chain guidance is not the only factor affecting difficulty, but it improves benchmark validity by filtering shortcut-solvable questions and making reasoning fidelity measurable.
>
> **Limited Scope of Source Data (W4)**: We acknowledge this domain limitation in Section 5. By extracting causal logic directly from scientific texts, our pipeline is domain-agnostic and scales to broader multimodal literature from PubMed Central.
>
> **References**:[1] Haozhen Gong, et al. Med-CMR: A fine-grained benchmark integrating visual evidence and clinical logic for medical complex multimodal reasoning. arXiv preprint arXiv:2512.00818, 2025.

---

> > ### Author Rebuttal · Reviewer_fsBs · 2026-04-01
> >
> > Thank you for the detailed rebuttal. It addresses most of my main concerns.
> >
> > My main remaining concern is Reference-Structure Bias. The current response usefully suggests that LCR is somewhat robust to differences in expression form, but “semantic fact verification” mainly addresses surface-form variation. The evidence is still not sufficient to fully establish systematic robustness to alternative but scientifically valid reasoning paths.
> >
> > Overall, the rebuttal has increased my confidence in the paper, and I plan to raise my score accordingly, toward a weak accept.

---

> > > ### Author Response · Authors · 2026-04-01
> > >
> > > Thank you for your constructive feedback and for considering raising your score. To address your remaining concern, we would like to clarify the design philosophy of our metric and how we will refine the manuscript:
> > >
> > > **1.Evidence-Constrained Reasoning**：
> > > - We explicitly posited in the introduction that the vast majority of scientific problems are fundamentally **evidence-constrained**, particularly in biomedical question answering (QA) [1, 2]. Within the context of pathology and biomedical VQA, the reasoning path must be strictly constrained by the specific visual evidence presented in the images [3]. Even if a model arrives at the correct conclusion, if its "alternative reasoning path" bypasses the actual visual phenomena provided in the context, it is highly likely that the model relies on semantic priors rather than the genuine extraction of visual evidence.
> > >
> > > - As emphasized by recent studies [2, 3], the reliability of clinical reasoning stems from a shift from surface-level linguistic fluency toward **evidence-grounded reasoning** and **reasoning soundness**. The LCR metric is specifically designed to verify this strict alignment between visual evidence and logical derivation. It ensures that the model's reasoning process is genuinely grounded in the provided data, thereby effectively identifying and penalizing "shortcut reasoning" that circumvents factual evidence.
> > >
> > > **2. Robustness and Alignment of the LCR Metric**：
> > >
> > > - **Empirical Validation against Shortcut Reasoning:** While the LCR metric may not encompass every atypical reasoning path, it statistically exposes the prevalent issue of LLMs utilizing "linguistic shortcuts" to bypass genuine evidence-based reasoning (empirically validated by Biology PhD annotators, with 90.1% of low LCR scores reflecting true factual errors rather than over-penalization). This confirms LCR's effectiveness in identifying the "false positives" systematically overlooked by outcome-oriented metrics.
> > >
> > > - **Exposing Deficiencies in RL Reward Designs**: Current RL paradigms for MLLMs often rely on outcome-based reward functions (e.g., final answer correctness), a practice that potentially reinforces flawed reasoning processes. Our evaluation reveals the inherent defects in past RL reward settings; specifically, models optimized via such outcome-oriented rewards (e.g., Lingshu, Fleming) perform poorly on SORBE, as they struggle with strict evidence constraints despite arriving at correct answers. A key contribution of LCR is exposing this systematic deficiency in current training objectives, highlighting the need for process-aware rewards.
> > >
> > > - **Scope and Inherent Limitations:** Admittedly, our evaluation protocol for open-ended question cannot guarantee absolute unbiasedness **(a prevailing challenge in the current evaluation of open-ended questions)**. On the other hand, it is able to statistically expose cases where models arrive at shortcut answers by bypassing the correct reasoning process, which substantially reduces evaluation error in practice. We will make this limitation explicit in the revised version.
> > >
> > > Thank you again for your rigorous review, which has greatly helped us clarify the specific scope and boundaries of our benchmark. We hope this fully addresses your final concern.
> > >
> > >
> > > **References**:
> > >
> > > [1] David L. Sackett, et al. Evidence based medicine: what it is and what it isn't. BMJ, 1996.
> > >
> > > [2] Yiqing Zhang, et al. PubMed Reasoner: Dynamic Reasoning-based Retrieval for Evidence-Grounded Biomedical Question Answering. arXiv preprint arXiv:2603.27335, 2026.
> > >
> > > [3] Haozhen Gong, et al. Med-CMR: A fine-grained benchmark integrating visual evidence and clinical logic for medical complex multimodal reasoning. arXiv preprint arXiv:2512.00818, 2025.

---

### Decision · Program_Chairs · 2026-04-30

**Decision:**

Accept (regular)

**Comment:**

This paper introduces SORBE, a benchmark for evaluating complex, evidence-grounded reasoning in biomedical VQA, along with a process-oriented metric (LCR) that jointly assesses reasoning steps and final conclusions. The work addresses an important limitation of existing benchmarks, which often focus on answer correctness rather than reasoning fidelity, and proposes a well-motivated and carefully designed evaluation framework. Empirical results provide valuable insights into the current limitations of state-of-the-art models in multi-step, evidence-based reasoning.

Reviewers generally agree that the paper addresses an important problem and makes a meaningful contribution to benchmark design and evaluation methodology. The proposed LCR metric and multi-image, evidence-centric setup are particularly appreciated. At the same time, several limitations remain, including concerns about domain diversity, potential evaluation biases, and the need for stronger validation of certain design choices.

The authors provided a thorough and constructive rebuttal, addressing many of the key concerns. In particular, they strengthened the evaluation by adding expert validation, improving the objectivity of the scoring mechanism, and clarifying the scope and generalizability of the benchmark. These revisions significantly improve the clarity and credibility of the work, although some limitations (e.g., domain scope and evaluation coverage) still remain.

Overall, considering the importance of the problem and the solid contribution, I recommend weak acceptance. I encourage the authors to incorporate the discussed revisions, especially clarifying the scope of claims and further strengthening validation, in the camera-ready version.